



# Re-discovering Robert E. Horton's Lake Evaporation Formulae: New Directions for Evaporation Physics

Solomon Vimal[1], Vijay P. Singh[2]

[1]Department of Geography, University of California, Los Angeles, CA, 90049, USA
[2]Department of Biological and Agricultural Engineering & Zachry Department of Civil and Environmental Engineering, Texas A&M University, College Station, Texas 77802-2117, USA
*Correspondence to*: Solomon Vimal (solomonvimal@ucla.edu)

**Abstract.**   Evaporation from open water is among the most rigorously studied problems in hydrology. Robert E. Horton, unbeknownst to most investigators on the subject, studied it in great detail by conducting experiments and heuristically relating his observations to physical laws. His work furthered known theories of lake evaporation, but it appears that it got dismissed as simply empirical. This is unfortunate, because Horton's century-old insights on the topic, which we summarize here, seem relevant for contemporary climate change-era problems. In re-discovering his overlooked lake evaporation works, in this paper
we: 1) examine his several publications in the period 1915-1944 and identify his theory sources for evaporation physics among scientists of the late 1800s; 2) illustrate his lake evaporation formulae which require several equations, tables, thresholds, and conditions based on physical factors and assumptions; and 3) assess his evaporation results over continental U.S., and analyse the performance of his formula in a subarctic Canadian catchment by comparing it with five other calibrated (aerodynamic and mass transfer) evaporation formulae of varying complexity. We find that Horton's method, due to its unique variable
vapor pressure deficit (VVPD) term, outperforms all other methods by ~3-15% of $R^2$ consistently across timescales (days to months), and an order of magnitude higher at sub-daily scales (we assessed up to 30 mins). Surprisingly, when his method uses input vapor pressure disaggregated from reanalysis data, it still outperforms other methods which use local measurements. This indicates that the vapor pressure deficit (VPD) term currently used in all other evaporation methods is not as good an independent control for lake evaporation as Horton's VVPD. Therefore, Horton's evaporation formula is held to be a major
improvement in lake evaporation theory which, in part, may: A) supplant or improve existing evaporation formulae including the aerodynamic part of the combination (Penman) method; B) point to new directions in lake evaporation physics as it leads to a "constant" and a non-dimensional ratio - the former is due to him, John Dalton (1802), and Gustav Schübler (1831), and the latter to him and Josef Stefan (1881); C) offer better insights behind the physics of the evaporation paradox (i.e. globally, decreasing trends in pan evaporation are unanimously observed, while the opposite is expected due to global warming).
Curiously, his rare observations of convective vapor plumes from lakes may also help explain the mythical origins of Greek deity Venus and the dancing Nereids.



## 1. Introduction

The problem of accurate lake or open water evaporation estimation has been a subject of scientific inquiry, in the modern sense of combined experimental and theoretical study, for the past four centuries. Factors that control evaporation have been investigated since the time of Edmund Halley (1687) with rapid progress in theories of thermodynamics, aerodynamics (turbulence theory), and molecular kinetics (kinetic theory of gases) that led to better understanding of evaporation due to wind's influence, convection, and diffusion. Brutsaert's treatise on "Evaporation Into the Air" provides an overview of concepts that evolved from antiquity (Brutsaert, 1982, Chapter 2). From the 1700s, key contributions have included those of Johann and Daniel Bernoulli (1700s); John Dalton, Rudolf Clausius, Osborne Reynolds (1800s); the celebrated voyage through turbulence theory (Davidson et al., 2011) from European, American, and Russian schools, among others, especially as data of field experiments on surface winds and diffusion became increasingly crucial for chemical warfare efforts over the course of the 20$^{th}$ century (Sutton, 1953). More recent developments include the recognition of the complementary principle of evaporation in the late 1900s (Bouchet, 1963; Morton, 1994; Brutsaert, 1982) and the evaporation paradox (Roderick and Farquhar, 2004) which have large implications in climate change debates.

Robert E. Horton, a pioneer in hydrology, well-regarded for his contributions to areas of hydrology like infiltration, overland flow, and river geomorphology, is not usually considered a fundamental contributor to the field of evaporation. However, unbeknownst to most in mainstream evaporation theory, tucked away in his home-based experimental catchment beside a pond, Horton conducted rigorous experiments and theoretical work on open water evaporation from the 1910s until the end of his career, circa 1945. In particular, in 1917 he published a set of formulae for estimating evaporation (including within lake variations of evaporation) based on physical laws which he believed were more robust than the then existing methods. The sub-text to the title of his first 1917 paper claims:

> "Empirical Statement Based on Physical Law Agrees with Observed Facts and Is Held To
> Be an Improvement Over Existing Formulas" – Horton (1917a)

He held the view that his equation was superior to other known methods for the following decades, even in the face of rapid developments in evaporation theory in that period (e.g. see Horton, 1934). After we examined several of Horton's papers and reports related to evaporation from lakes and pan evaporimeters (or simply, pans) from 1917 to 1944 (the year before his death), we noted that he derived his formula theoretically, but since the values of the coefficient in his formula were not easily available, and his formula resembles other empirically derived formulae, several investigators may have dubbed it as simply empirical (see Rohwer, 1931). However, Horton's nuanced understanding of the boundary layer physics of his time (turbulence theory, horizontal vapor transport via laminar flow, convective transfer of vapor, wind and vapor blanket characteristics), and the sound premise of his work based on molecular kinetics, reveal the potential of his work to offer new insights for an improved formulation of evaporation. The theory behind his work is illustrated in Sect. 2. After evaluating Horton's evaporation formulae (in Sect. 3), we find that his claim of having developed an improved method not only stands



to be true in his time, but also holds great contemporary value, and it is unfortunate that it has been largely overlooked or
forgotten. Therefore, in this paper we examine his evaporation work from the perspective of contemporary theories as well as
those of his time to highlight his ingenious perceptual, experimental, and theoretical insights into the subject. We revisit his
claims, replot his figures with recent data, simplify the use of his experimental tables (by converting them to parametric forms),
assess his method's ability to generalize across wide-ranging conditions, and show the relevance of his method for
contemporary large-scale evaporation problems.

### 1.1.  Horton's broader contributions and bibliography


Hydrologists need no introduction to some of Horton's contributions like infiltration theory, overland flow,
geomorphological laws, etc. What may not be widely known is that he published an estimated 200 papers and reports, mostly
single authored (~90%), but of these, only about 80 works are available from readily accessible sources (Hall, 1987). Horton's
unpublished works are held at the U.S. National Archives in College Park, Maryland (cataloguing and organization was done
by Walter Langbein). A subset of his archive is also held at his alma mater Albion College (Accavitti, 2019). In the last few
decades, Dr. Keith Beven from Lancaster University and Dr. Jim Smith from Princeton University examined a portion of the
archive contents and presented their findings via publications (Beven, 2004 a, b, c) and an AMS Horton lecture (Smith, 2010).

About 80 of Horton's contributions were provided by Hall (1987) and curated by the AGU Virtual Hydrology Project
(see Foufoula-Georgiou, accessed 2021-05-19). A more complete list of Horton's works was collated by Dr. Elizabeth Clark,
which includes ~135 works, for an American Meteorological Society (AMS) Horton Lecture delivered by Dr. Dennis
Lettenmaier (Lettenmaier, 2008). Combining these lists and conducting additional searches, the first author collated 168
works, the most comprehensive list of Horton's works available to our knowledge. Years and titles are shared in
Supplementary.

### 1.2.  Horton's lake evaporation method and related projects


About a dozen of Horton's papers and reports are related to his evaporation method and supporting ideas, but one
can get a full understanding of his published contributions on lake evaporation from four key publications: Horton (1917a,
1927, 1934 and 1943b). Horton's evaporation method was first introduced in Horton (1917a), as part of a three-paper series
(Horton 1917a, b and c) in *Engineering News-Record* for the purpose of improving waterpower, water-supply and irrigation
projects. The larger goal of the three papers was to reduce errors in estimates of stream yield, especially to get accurate
estimates of low flows to ensure the success of hydraulic (water supply) projects. This goal necessitated reliable evaporation
estimates, leading Horton to developing his own method to calculate it. The tables needed to implement his method were not
published in entirety in Horton (1917a), but only in a later report on Great Lakes a decade later (Horton, 1927) which was a
major project in his career involving a rigorous procedure for lake evaporation estimation among a broader hydrological study



of the Great Lakes. This work was conducted in collaboration with C.E. Grunsky, and was an extensive 432-page report. The
central innovation of this contribution is that prior to this work, it was not possible to achieve correlations between discharge
and lake levels which are impacted by a variety of natural and artificial causes. A substantial portion of the report is a
presentation of available data related to the hydrology of the Great Lakes and the remaining is an analysis of various aspects
of the water balance (precipitation, runoff, evaporation) including 142 tables and 73 figures. In another paper 7 years later,
Horton (1934) provided more theoretical insights into his evaporation method with some explanation of its physical basis.
Besides these major works on the evaporation method itself, projects where he discussed or estimated lake evaporation
spanned earlier and later times in his career: e.g. in Horton (1905), he discusses evaporation and water balance in the context
of small kettle ponds, and in Horton (1944), he did so in the context of a dam design for the Hemlock lake system. As a final
point to contextualize his lake projects, it is worth noting here that Horton's experimental catchment beside his house included
a pond about 200 meter long and 60 meter wide (a figure is provided in Horton, 1919a) where he conducted some key
evaporation experiments and interesting observations (we revisit this in Sec. 7, Closing Note).

### 1.3. Previous examinations of Horton's lake evaporation method

The various above-mentioned works related to lake evaporation have been cited sparingly which shows that they
were largely overlooked. They have not been collectively examined in any previous work to our knowledge, and in the few
citations to them, the value and sophistication of the method was not recognized. Horton's lake evaporation equation received
some attention in Chow's Handbook (in Sect. 11 on evaporation written by F. J. Veihmeyer; Chow, 1964). Horton's formula
is surprisingly not included in Brutsaert's treatise (Brutsaert, 1982) which has ~650 citations of evaporation-related works,
though his work on evaporation pans has been cited, referencing standardized class-A pans. The equation was cursorily
reviewed in a few recent studies. McMahon et al (2016) cite the equation (presumably taken from Chow, 1964) as part of a
larger review together with other evaporation equations. Singh and Xu (1997) evaluated Horton's evaporation equation in
comparison with 12 other (mostly) empirical equations that resemble it, but incorrectly, in the sense that the vapor pressure
deficit (VPD) was multiplied with the wind factor, whereas for correct use of Horton's equation the wind factor is to be
multiplied with vapor pressure of water, and not the total deficit - a fundamental difference between his method and other
methods (as will be explained in Sections 2 & 3 in more detail). As inferred from citations to his key evaporation paper
(Horton, 1917a) via Google Scholar (accessed, April 29, 2021), few investigators from Russia and Portugal have examined
his evaporation work, and one particular work from Japan (Siomi and Yosida, 1940) seems to have examined Horton's
equation in some detail, but not as comprehensively as we undertake here. All these works do not account for the full
complexity of his approach: for comprehensive use of Horton's lake evaporation method, about 20 equations and two tables
are needed (Sections 2 & 3). One of these tables was not very accessible, as it was published in a report (Horton, 1927) which



presumably was not so widely circulated as an academic journal, which we believe may have led to the limited use of his
method.

### 1.4. Mainstream evaporation works in Horton's time

For the context of works preceding Horton's time, interested readers are ushered to an excellent contribution by Grace Livingston published as 8 pieces in *Monthly Weather Review* between 1908 and 1909 and later compiled into a book (see Livingstone, 1910). This annotated bibliography includes ~850 works on evaporation from late 1600s up to the early
1900s, lists 155 publication outlets, and was translated from multiple world languages (Japanese, French, Italian, German, Russian, among others). It is possible that Horton considered his equation as an improvement over other evaporation formulae presented in this review. Horton did not cite this bibliography in any of his evaporation papers, but there are multiple reasons to speculate why he might have examined it: 1) many of Horton's works were published in the same journal (*Monthly Weather Review*); 2) he followed an unconventional citation style and often included no reference lists in his papers (e.g. see Horton,
1917a); 3) Mrs. Grace Livingstone was the ex-wife of a plant physiologist, Burton E. Livingston, whose work on evaporation Horton certainly followed (Horton, 1927); 4) the compiled book format of the annotated bibliography (Livingstone, 1910) was available at the Weather Bureau Library in Washington and John Crerar Library in Chicago, places that Horton presumably frequented due to their proximity to the work he did in Chicago and his engagements with members and initiatives of the Weather Bureau (Horton, 1927); and finally, 5) most, if not all, of the theoretical sources that Horton's evaporation
method relied on (discussed later in see Sec. 1.5) appear in one place in Livingstone (1910).

Horton's evaporation method was apparently developed and used in New York, Michigan and Chicago (see Horton, 1927), but in the same time period many similar efforts were underway throughout the United States (presumably in other countries too). Three such works are worth highlighting: 1) The thermodynamic approach using Le Chatelier's principle applied to energetics was undertaken in California at the *Scripps Institute of Oceanography* and *California Institute of*
*Technology*, which led to the energy balance solution of lake evaporation, and the Bowen ratio (Bowen, 1926). Subsequent works by others that picked up on this work are summarized in a succinct compendium by McEwens (1930) and a historical summary by Lewis (1995). 2) A review of mass transfer and energy balance based evaporation studies on Lake Hefner resulting from collaboration between several *U.S. agencies: Geological Survey*, *Department of Navy*, *Bureau of Ships*, *Navy Electronics Laboratory*, *Department of Interior*, *Bureau of Reclamation*, *Department of Commerce*, and *Weather Bureau*
(USGS, 1954). 3) A statistical attack on the problem led by geophysicist J. F. Hayford, who notably spent over 2000 hours developing a superior method, including a mammoth effort by 41 persons who collectively spent some 32,000 man hours on this work (Folse, 1929, p. 7). The method uses temperature and humidity of the preceding day to calculate the following day's evaporation, and includes a large system of equations with many free parameters, which is optimized to minimize error (for more details see Folse, 1929). It was developed for the Great Lakes, and did perform reasonably well there, but generalized



poorly in other lakes, and did not gain wider attention (see critical review by Bernard, 1936). These highlight some the various independent efforts dedicated to calculating evaporation around the time when Horton's method was developed.

### 1.5. Horton's main sources for theories and experiments of lake evaporation physics

Citations provided in Horton's work show that he relied on the works of several European scientists for concepts related to the physics of evaporation. He did examine several empirical equations developed in the US (see Horton, 1934),
but he does not appear to have followed the works conducted by Bowen and Cummings (Bowen, 1926). Perhaps this is because Bowen's works appeared in *Physical Review*, while Horton published his works in *Monthly Weather Review*. Moreover, Horton's approach differed in that it was premised on aero-hydrodynamics and kinetic theory approaches which were developed mainly by European scientists.

A molecular kinetics view of evaporation is fundamental to his approach, and he developed this view mainly from
John Dalton's theories and experiments on evaporation of water and other chemicals (Dalton, 1802). Dalton's work was in fact the only work that he directly cited when he first published his evaporation paper (Horton, 1917a), though with a closer look through his later papers (Horton, 1927 and 1933), it does appear that he developed his method by building upon multiple works. It appears that he studied: Thomas Stevenson's (1882) work on wind speed variation by height, while conducting his own experiments on the role of wind on evaporation (see Horton, 1927); Geoffrey I. Taylor (1918) for the role of turbulence
and vapor blanket (Horton, 1934); Napier Shaw's *Manual of Meteorology* (Shaw, 1932) and Julius von Hann's *Lehrbuch Der Meteorologie* (von Hann, 1926) for work on Psychrometry (see Horton, 1934, and also Horton, 1921, though no citations are provided in the latter); Thomas Tate (1862) for laws of evaporation; Josef Stefan (1881) for water surface's geometric controls on evaporation and also perhaps the role of vapor blanket in turbulent and convective transfer of vapor from large and small water bodies. Stefan is cited in Horton (1934), but Stefan's work may have also inspired his equations in Horton (1917a) due
to their resemblance. A reference to a Chemistry book he read in his youth (from his short story collection, see Horton, 1938) can be traced to *"A Dictionary of Chemistry"* by James Watts (Watts, 1882) wherefrom Horton learned about a sampling method to collect combustible marsh gases from shallow ponds and lakes. In a posthumous work on convectional vortex rings (Horton and van Vliet, 1949), he uses P. G. Tait's acid experiment to understand convection (Tait lecture, 1878, referenced in Dolbear, 1894 and Risteen, 1896) which gives one a mental picture of how he viewed convective evaporation from lakes.
From these references, we can see how his Physical Chemistry knowledge developed over the course of his life.

His references also included American textbooks, two in particular: Allen Risteen's *Molecules and Molecular Theory* (Risteen, 1896) and Amos Emerson Dolbear's *"Matter, Ether and Motion"* (1892). Risteen's work is cited in Horton (1934) where his evaporation formula is discussed in more detail than in previous papers. It appears that Horton's collaborator van Vliet, who published Horton's work on convectional vortex rings posthumously (Horton, 1949), misspelled his reference to
Dolbear as Dalhaer (perhaps a transcription error). These American textbooks referred to theories developed in Europe by





Rudolf Clausius and a treatise on Kinetic Theory of Gases (Watson, 1876). Watson's work on kinetic theory, in turn, credits the origin of these theories to Johann Bernoulli, James Clerk Maxwell, Rudolf Clausius and Ludwig Boltzmann. Most of these scientists were aerodynamicists, physicists, and chemists. Notably, Dolbear was not only a physicist but also a pioneering inventor who competed with Alexander Graham Bell at the Supreme Court of the U.S. for priority on the patent of the

telephone (his claim was that he invented it 10 years earlier, but he lost the case). Nearly all of these books are available for free from Google Books (full reference and hyperlinks are provided in the reference list).

## 2.   Premise of Horton's evaporation formula

Before we delve into the details of the evaporation equation, a quote from Horton contextualizes how he supposedly viewed his evaporation formula:

> "A rational equation may be defined as one which can be derived directly from fundamental principles, which fits all the experimental data and which represents the physical conditions correctly throughout the entire range of their occurrence and hence is valid outside the range of experimental observation" – Horton (1941).

Some fundamental principles he alluded to in his evaporation formula are related to thermodynamics (i.e. work done

in phase changes, latent heat), and they include references to geometric proofs of the same from the perspective of kinetic theory drawn from Risteen (1896), discussed in Horton (1934). More importantly, the premise of Horton's fundamental principles in his evaporation method is the kinetic theory of gases (Loeb, 1934) which he explicitly stated in Horton (1917a). His molecular kinetics view of evaporation is best captured in the following quote:

> "In a mixture of air and water-vapor there is a certain number of vapor molecules per unit volume. When there is wind the air and vapor are swept along together at a rate depending on the pressure-gradient. This, as in case of hydraulic flow, is independent of the total pressure. At a given vapor-pressure the same amount of vapor is carried by the wind per unit of time and per unit of volume of air, whether the number of air molecules per unit volume is large or small." – Horton (1934).

Horton considered the movement of molecules and their behavior at the surface of the lake as three key processes: 1) vapor emission, 2) vapor removal (by diffusion, convection, and wind action), and 3) vapor return. These processes are discussed in multiple papers (Horton 1917a and 1934). It may benefit the reader to review these three processes in some detail before introducing the evaporation equations.

### 2.1.   Vapor emission and vapor return

His first paper on evaporation (Horton, 1917a) does not discuss the thermodynamic perspective, but his derivation of the various parts of the evaporation equation does use the underlying principles, as exemplified in the following quote:

> "[Latent heat] comprises of two elements: (1) Internal work in overcoming molecular attractive forces which, in general, including viscosity and surface-tension, increase as the





temperature decreases, and the latent heat of internal work also increases as the temperature decreases; (2) the external latent heat, which measures the work done by the emitted vapor in expanding against the external pressure, decreases slightly as the pressure on the liquid surface decreases with decreased boiling temperature, but the total latent heat increases slowly as the temperature decreases." – Horton (1934)

He examined these thermodynamic factors to identify the role of pressure in impacting vapor emission and vapor

removal. While pressure does affect vapor emission rates due to external latent heat, it is negligible, so the impact of pressure

on evaporation can be attributed to vapor removal (somewhat like a proof by elimination).

Vapor return is controlled by wind action (which is non-linear) and the vapor pressure of the overlying air or the

*vapor blanket*, i.e. a thin layer of vapor just above the water surface analogous to viscous sub-layer in open channel flow. The

characteristics and role of vapor blanket is discussed separately in more detail in Sec 3.4.

**2.2. Vapor removal**

Vapor removal, as previous stated, happens due to diffusion, wind action and convection.

**2.2.1. Diffusion**

Horton's conception of evaporation via diffusion is perhaps drawn from Dalton's (1802) original work which is the

only reference he cites when he first published his lake evaporation formula in Horton (1917a). Dalton posited:

"Evaporation [...] is caused by *vis inertiae* of the particles of air; and is similar to that which a stream of water meets with in descending amongst pebbles […]. From a great variety of experiments [on evaporation,] I have found the results entirely conformable with the above theory […] – Dalton (1802, pp. 581-584).

The rate of diffusion is governed by water temperature (for vapor emission rate) and barometric pressure and vapor

pressure of air (vapor return rate), and is not explicitly affected by wind action or convection (Horton, 1934).

**2.2.2. Wind action**

According to contemporary evaporation literature (see Brutsaert, 1982), wind can have two effects: 1) turbulence

transfer of vapor away from surface; and 2) advective (bulk fluid mass) transport due to mean horizontal wind. In Horton's

work, wind action is considered separately as a bulk exhaustion process that removes vapor at a maximum rate equal to the

rate of vapor emission. The rate of wind action in Horton's work is based on Dalton's observation:

"[Dalton] found that a strong wind made the amount of evaporation *double* that taking place in still air. He concluded that the increase in evaporation rate was proportional to the wind velocity" – Horton (1917a)


Evaporation by horizontal advection seems to be included in Horton's conceptualization of wind action (it is
considered indirectly), where for a given elemental area, the vapor pressure of water is amplified by the wind up to a limiting
value, which indirectly accounts for the rate of vapor removal by advection and turbulent transfer: they are not differentiated.

### 2.2.3. Convection

It may help the reader to first disambiguate the term convection as it is sometimes used interchangeably with
advection (e.g. convection-dispersion equation/advection-dispersion equation). Convection normally refers to heat transport
via vertical plumes in fluids when wind shear is overcome by thermally driven buoyant production of kinetic energy, while
advection normally refers to transport of quantities (heat or matter) due to mean horizontal flow of wind (see Hess, 1979;
Stull, 1988; and Eagleson, 1990). Horton's usage of the term convection does share similarities with the common parlance in
turbulence theory pertaining to heat transport, i.e. convection happens due to expansion from surface air heating as well as
vapor addition which causes a reduction in density (as the bulk air is heavier than moist air) which result in instability.
Convective plumes are fed and sustained by laminar wind that feeds moisture horizontally into it, and continues until the
buoyant force overcomes the shear force due to horizontal wind. It is sustained until the moisture available to feed the plume
is depleted. This conceptualization of convection is not clearly described in Horton's evaporation papers, but we inferred it
from the following quote in his paper (Horton, 1933) on columnar vapor drift (a mechanism of evaporation):

> "In the eerie morning hours [...] vapor columns present a spectral appearance as they travel
slowly over the water surface, resembling sheeted ghosts or white-robed whirling
> Dervishes walking on the water. […] Obviously columnar vapor drift [also amorphous
> vapor drift] is a visualization of convective vapor removal from a water surface during
> evaporation. [...] A vapor column forms wherever a sufficient degree of instability develops
> through the warming of a layer of air close to the water surface and through the
accumulation of water vapor (which is lighter than air) therein. A vapor column is fed by
> horizontal flow of air and vapor toward it close to the water surface. Apparently it grows
> until its feeding area encounters another area from which the vapor has already been
> exhausted or until the frictional resistance of horizontal flow balances the vertical
> convective forces" – Horton (1933)

Horton regarded convection as a *rheologic* system, i.e. a flow process with solid and fluid characteristics, typically
in response to forces (in the case of evaporation, as pressure over a unit elemental area). In the following quote, his view of
convection as a rheologic system is clearly stated:

> "The ordinary, vertically convective system […] may be considered hydrodynamically as
> a rheologic or flow system, resembling the flow through a vertical pipe connecting two
reservoirs, with lower pressure in the upper reservoir. This may be called the tubular type
> of vertical convection." – Horton (1949)



While numerous physical factors were taken into consideration in his understanding of evaporation, to get a mental picture of Horton's conceptualization of processes that govern evaporation, the schematic below (Fig. 1) may serve as a graphical summary of the key processes related to evaporation.

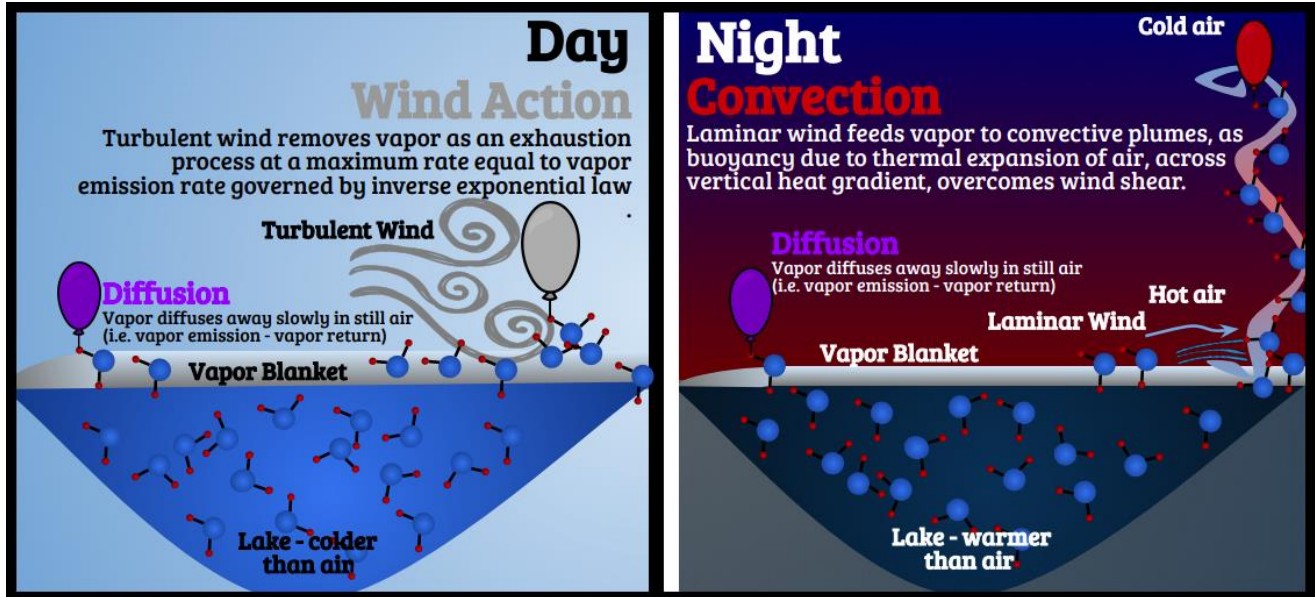


**Figure 1: Horton's understanding of primary processes that control evaporation.**

Here, the colored balloons represent evaporation aided by vapor removal due to diffusion (purple), wind action (grey) and convection (red). Diffusion can be upward or downward in direction: upward (positive) is evaporation and downward (negative) is condensation (see Sec. 3.5). Grey balloon (wind action) depends on wind speed about 1 foot away from the
surface of the water, and it is governed by an inverse exponential law (see Sec. 3.2) and can happen during day or night, though it is accentuated during the day when wind speed is higher. Red balloon (convection) depends on temperature deficit across a vertical gradient and laminar wind that accompanies vapor removal (see Sec. 3.2.3) and it occurs predominantly during the night (Horton, 1917a) when water is warmer than air due to its higher heat memory (i.e. specific heat capacity).

## 3. Illustration of Horton's evaporation method

In what follows, we illustrate Horton's evaporation equations, their theoretical basis (where possible using direct quotes), correction factors and tables (as parametric equations) and provisional values of coefficients with appropriate units.



### 3.1. Evaporation equations: pan evaporation, evaporative capacity and lake evaporation

If $V_w$ is the saturated vapor pressure at surface water temperature ($\theta_w$) and $v_a$ is the actual vapor pressure of overlying air a small distance above the water surface at air temperature ($\theta_a$), the Dalton Factor (more commonly called the vapor

pressure deficit, **VPD**) is $[V_w - v_a]$. All evaporation equations use VPD, but in Horton's equation for evaporation, the VPD term is replaced with a **variable VPD** term (**VVPD**), $[\Psi V_w - v_a]$, where the variable $\Psi$ is called the *wind factor* (elaborated in Sec 3.2). $\Psi$ is not to be confused with a constant factor: it varies with meteorological conditions and has no units. Its values range from 1-2 ($1 \leq \Psi \leq 2$) depending on near-ground wind speed ($w_0$), to account for vapor removal by wind action and convection from the vapor blanket (discussed in Sec. 3.2.4). There are multiple reasons behind the position of $\Psi$ in VVPD

which can be inferred from Horton's papers (1917a, 1927 and 1934). We discuss these reasons in Sec. 3.2. with direct quotes to Horton, where appropriate, to convey his thinking.

*Pan evaporation* ($E_P$), which is the same as *evaporative capacity from lake ($E_{Cw}$)*, is assumed by Horton as

$$E_P = C \underbrace{[\Psi V_w - v_a]}_{VVPD} \tag{1a}$$

$C$ is a constant related to time and elemental area over which evaporation happens and the units of measurement of

evaporation and vapor pressure. He measured vapor pressure in inches of mercury and wind speed in miles per hour. The provisional values he prescribed for $C$ (in inch per time units) are: 0.4 for a small elemental area, 0.36 for a 12 square inch pan over daily scale, 12.2 for an average month of 30.42 days, 73.2 for 6 months. Some of these provisional values for C are given in Horton (1917a) and others in Horton (1927). According to Horton (1917a), these values are not standardized and are subject to revision. We provide revised values in Sec. 3 (Table 3).

*Evaporation capacity ($E_C$)* relative to air is calculated w.r.t. saturated vapor pressure of air ($V_a$) as

$$E_C = C[\Psi V_a - v_a] \tag{1b}$$

He defined $E_C$ as:

> "The maximum rate of evaporation which can be produced by a given atmospheric
> environment from a unit area of wet surface exposed parallel with the wind, the surface
> 320 having at all times a temperature exactly equal to that of the surrounding air." – Horton
> (1919a)

For small water bodies, particularly those with shallow depth, in the absence of water surface temperature data, when the lag between water and air temperature is negligible, Eqn. (1b) can be used. Over pans, an area factor and the variability of vapor blanket thickness should be taken into account (discussed in Sections 3.3 and 3.4), but can be ignored over large lakes.

*Lake evaporation ($E_L$)* is calculated w.r.t. vapor pressure of overlying vapor blanket ($V_b$) as

$$E_L = C[\Psi V_w - V_b] \tag{1c}$$

Vapor pressure of vapor blanket ($V_b$) is calculated from the corresponding vapor blanket temperature, $\theta_b$ (Horton, 1927, pp.161) using what is now called the Clausius-Clapeyron relationship, but in Horton's time this was calculated using





graphical Psychrometric charts (see Horton, 1921). Vapor blanket temperature is approximated by a simple relationship, $\theta_b = \theta_w + (\Psi - 1)\Delta$, where $\Delta$ represents the difference between surface water and air temperature regardless of sign, i.e.: $\Delta = |\theta_w - \theta_a|$, where $\theta_a$ is air temperature. The expression for $\theta_b$ appears to be only a heuristic (i.e. an approximation with no theoretical basis) that may be applicable only in monthly time scales. Furthermore, Horton (1927, pp. 161-162) noted that it works for small variations of $\theta_w$ from $\theta_a$, but suggested that if the air temperature is much higher than water, when relative humidity approaches 100%, then the relationship may not hold, because under such a condition, the distance over which vapor blanket becomes fully formed approaches infinity (see Sec. 3.4).

### 3.2. Wind Factor ($\Psi$)

The inclusion of $\Psi$ in the VVPD terms is what leads Horton's equation to generalize across a variety of physical conditions and perform better that several other equations (see Sect. 4), and what makes us consider Horton's evaporation formulae semi-empirical or quasi-physical (or "rational" in Horton's terms, see Horton, 1941).

The wind factor, $\Psi$, depends on the wind velocity close to the water surface ($w_0$) which, when convection is ignored, is assumed to be of the form of an inverse exponential law

$$\Psi = \mathcal{H} - e^{-kw_0} \tag{2a}$$

In this paper, $\mathcal{H}$ is designated as the Horton lake evaporation constant. Horton assigned it a constant value of 2 but it could be a little lower (discussed in Sec. 4). For the value of k, a constant called the *wind coefficient*, Horton prescribes values of 0.2 or 0.3 depending on the exposure of the evaporation pan (Horton, 1917a), but our experiments (as will be shown later, see Table 3) show it can be as low as 0.13. Apparently, $\Psi$ values change depending on the values assumed for k and the $\Psi$ tables Horton published (provided later as parametric equations in Sec. 3.2.4) are for k=0.3 (Horton, 1943b).

**Adjustment of $\Psi$ for convective vapor removal in light (or absent) wind:** In the case where warm days are followed by cool nights, convective vapor removal may be important. Convective vapor removal happens more readily in the night times than in the day times. When surface winds are suppressed by inversion, and when water temperature is higher than that of air, evaporation may be dominated by convection, so an alteration of the formula for $\Psi$ given by Eqn. (2a) is required. Horton's observations suggest that for ordinary natural temperatures, the $w_0$ in exponent can be replaced by the $w_0 + \sqrt{\theta - \theta_a}$, which would then include the effect of convective transport in the absence of strong winds (given below as Eqn. 2b). To calculate the combined convection and wind action when wind speed is low, conditions where convection prevails can be related to a Beaufort force scale for light or calm. Horton does not specify a threshold, but he prescribes 2 mph in an example problem. Therefore, when convection is not ignored, under mild winds, when $\theta > \theta_a$, $\Psi$ under these conditions is given by

$$\Psi = \mathcal{H} - e^{-k(w_0 + \sqrt{\theta - \theta_a})} \tag{2b}$$

where $\theta$ and $\theta_a$ are temperatures of water and air measured in Fahrenheit.





**Theoretical basis of Ψ in relation to physically-based methods:** One familiar with the combined equation of Penman may
recognize that Horton's approach to adjust the wind term with a convective term bears some resemblance to the physics
represented in the combined equation which uses a harmonic mean-like weighting, wherein the psychrometric constant
accounts for the role of pressure (the aerodynamic term) and slope of the saturation vapor pressure curve accounts for the role
of temperature (the energetics term) together forming the combination method. Similarly, Horton's assumption that convection
is caused by a combined effect of calm wind and temperature gradient appears to be logically related to part of the physics
represented by the *Flux Richardson Number ($R_{if}$ = -B/P)*, i.e. the ratio of buoyancy production (B), which represents buoyant
force from vertical temperature gradient (turbulent heat flux), to that of shear production (P), an aerodynamic term (momentum
flux times wind velocity gradient). Refer to Stull (1988) and Hess (1979) for their derivations. Understanding these
relationships may lead to improved formulations of Ψ.

### 3.2.1. Assumptions behind Ψ and rationale for its position in VVPD

Though the rationale behind Ψ is not discussed in his first paper where the evaporation method was introduced
(Horton, 1917a), in the context of applying his equation under varying conditions of pressure (elevation), in a paper 17 years
later (Horton,1934), Horton clarifies the main assumptions behind the usage of Ψ and the rationale for its position in VVPD,
which can be summarized as four key points: A) non-linear control of wind; B) wind as an exhaustion process; C) upper limit
of wind's influence; and D) wind's influence on condensation. As these are the main reason for the superior performance of
his method, we discuss them briefly with direct quotes where applicable.

**A) Non-linear control of wind:** This assumption is motivated by a simple physical reason, apparently not considered
elsewhere by the numerous other investigators who studied evaporation by the mass transfer mechanism:

> "Most existing evaporation formulas are in error in that they involve a linear factor for
> wind correction such that wind effect apparently increases indefinitely as the wind velocity
> increases. It has been proved experimentally, and is indicated by physical considerations,
> that since the wind can do no more than to remove the water vapor as fast as it is emitted
> from the liquid surface, there is a maximum or limiting value of the wind factor
> corresponding to each water surface temperature." – Horton (1917a)

Other investigators followed Dalton's suggestion and included a wind correction factor that assumes the form
$f(u) = (1 + Kw)$ where the wind velocity *w* is multiplied by a factor *K*. Further, equations of this type do not account for
Dalton's important observation that evaporation doubles with strong wind:

> "with the same evaporating force, a strong wind will double the effect produced in a still
> atmosphere." – Dalton (1802, see pp. 581-584).

The value of 2 for $\mathcal{H}$ in Ψ can therefore be credited to Dalton's experiments on evaporation, but it was also verified
by Horton's own experiments with wind under varied conditions (Horton, 1917a).



**B) Wind as an exhaustion process:** To our knowledge, wind's role on vapor removal as an exhaustion process has not been studied by other investigators.

> "The removal of vapor by wind corresponds to a condition of natural exhaustion to which the inverse exponential law commonly applies." – Horton (1917a)


The theoretical basis for such a view appears in some detail in Horton (1934):

> "$\Psi$ [is] a wind-factor, based on the assumption that mechanical removal of vapor by the wind is of the nature of an exhaustion process and hence follows the inverse exponential or inverse compound interest law. It is also based on the assumption that the maximum possible effect of wind-action is to remove the newly emitted vapor from contiguity with the water-surface as fast as it is emitted." – Horton (1934)


"Natural exhaustion" mentioned in this quote is analogous to Horton's use of natural exhaustion in his paper on the physical interpretation of infiltration excess (see Horton, 1941), where he explains that its physical basis can in part be justified from first principles, and such use of inverse exponential law is at least "semi-rational" (quasi-physical), as it gives a complete

picture of the physical characteristics (in this case evaporation) under natural conditions. Based on the physics described by Horton, we infer that natural exhaustion happens from the reservoir (vapor blanket) of saturated vapor that is replenished by the vapor pressure of the water surface, which is then depleted by wind action and convection. Multiplying $\Psi$ with the total vapor pressure deficit (or the vapor pressure of air) would not represent the same. This point will become clearer in Sec 3.5 where the constituents of the evaporation formula are discussed.

**C) Upper limit of wind's influence:** Horton provides a rational basis for the upper limit of $\Psi$ in the following quote:

> "In accordance with the Dalton formula, with the form of wind factor hitherto commonly used, the rate of evaporation increases indefinitely as the wind velocity is increased. This is obviously incorrect, since the rate of evaporation cannot in any event exceed the rate of vapor emission, and the latter is not affected by wind velocity in the absence of waves and spray. These must be for each water-surface temperature a maximum rate of evaporation, which rate cannot be increased by further increase in the wind velocity" – Horton (1917a)


The rationale for the wind factor can be understood by considering the extremes: when evaporation is at its maximum rate, when wind speed is high (i.e. evaporation happens at double the rate as compared to still air, as Dalton observed), i.e. $\Psi = 2$, the formula for evaporation reduces to $2CV$, assuming $v=0$ (i.e. the air is fully dry) since we are interested in the extreme

case. In the other extreme, if wind speed is 0 and humidity is high, $\Psi = 1$, so Horton's equation reduces to free diffusion in still air, similar to Dalton's equation, $C(V-v)$.

The limitations of Dalton's evaporation work were well-known before Horton's time. For example, it has been noted that Dalton's observations were for the month of August only, and evaporation estimated using his equation were found to be imprecise in other Summer months (Soldner 1807). Also, Dalton's observation of doubling of evaporation rate in strong winds

has had further refinements in other studies of Dalton's time, one of which is mentioned in Brutsaert (1982):



"[Soldner's] perceptive remarks notwithstanding, during the next half century, apparently little progress was made as regards the effect of the air stream. […] Schübler's [1831] data obtained during 1826 at Tübingen […] showed that evaporation of a water surface exposed to wind was 1.7 times larger than that of a sheltered surface in summer, and 4 times larger in winter." – Brutsaert (1982)

Other studies in Horton's time independently allude to similar results from experimental observations (e.g. see Kennedy, 1933). Some insightful observations by these various works by Dalton, Schübler, Soldner have not been taken into consideration in modern mass-transfer formulations of evaporation. Going by cited references, it appears that Horton's work happened independently from Schübler's, Soldner's, and Kennedy's, while it did build upon Dalton's observations.

**D) Wind's influence on condensation:** Besides evaporation, diffusion and convection, and pressure effects (discussed later in Sec. 3.5), Horton's equation is robust to condensation. His equation's ability to generalize for condensation is another distinct and physically meaningful feature that differentiates it from other Dalton-type empirical equations, and also motivates the position of wind factor $\Psi$. Horton (1917a) writes,

"Condensation or dew rarely occurs on windy nights […] experiments were made to determine the effect of wind on the condensation of moisture on the surface of cans containing ice and water, and mixtures of ice and salt"

In a paper 17 years later, Horton (1934) revisits the role of condensation, revisiting experimental results in conjunction with the properties of his equation, and he writes,

"It is evident that wind—except a slight wind—does not affect the rate of vapor-emission and return by diffusion but it does increase the rate of mechanical removal of newly emitted vapor. Consequently it appears that wind tends to decrease condensation instead of increasing it."

These observations agree with Rohwer's (1931) experiments which Horton (1934) verified. Kennedy (1933) observed that when water is cooler than air, and for humidity above 77 per cent, condensation occurs under such sub-adiabatic conditions, but Horton (1917a) nuances this further, adding that condensation happens only under low windspeeds, and decreases with increasing wind speed, which apparently is captured with the formulation of VVPD.

### 3.2.2. Adjustment of $\Psi$ for pan geometry

Lake evaporation calculation is not contingent on the availability of pan evaporation data, but pan evaporation (as shown in Sec. 2.7) can be used to cross-check actual lake evaporation. The wind speed at ground has to be corrected considering the pan diameter (D) and depth (d) below the rim and a factor $\rho = \frac{10d}{D}$. Pan evaporation is calculated as

$$\Psi = \mathcal{H} - e^{-k(w-\rho)} \tag{2c}$$

The use of pan data as a proxy for lake evaporation is justified after due consideration of various factors that cause lake and pan evaporation to differ from each other, namely: 1) humidity corrections, 2) rim height and depth effects, and 3)





vapor blanket formation and exhaustion characteristics governed by meteorological factors (wind speed); 4) temperature

difference between pan and lake surface.

Horton felt quite strongly about improper usage of pan data, especially when they are land-exposed as it appears
from this quote:

> "The land-exposed evaporation pan appears to be about the poorest device humanly
> contrivable for the purpose of determining the evaporation losses from broad water
> surfaces." - Horton (1917a)


### 3.2.3. Values of Ψ and ground wind velocity

Horton (1927) conducted ingenious experiments on wind that circumvented the need for wind tunnels:

> "For the purpose of determining the effect of wind on evaporation, experiments were
> carried out at the author's laboratory, using pails filled close to the rim, and suspended so
> 470  as to swing freely from a rotating frame.[...] These experiments and studies served to
> determine the coefficients in the formula." – Horton (1927)

Wind factor ($\Psi$) changes based on wind speed measured near the ground ($w_0$). He calculated $w_0$ based on his and
Stevenson's experiments for velocity variation by height (see Stevenson, 1882), but he only published the data in a report ten
years after the publication of his equation. The table provided by Horton (1927) for $\Psi$ can be converted into a cubic polynomial

with coefficients that have 5 decimal places for values of wind speed ranging from 0-15 miles per hour (mph), or equivalently
0-6.7 meter per second (mps). For wind speeds beyond this limit, the value of $\Psi$ can be linearly interpolated between 1.95
and 2 as a reasonable approximation. However, at near ground level (at about 1 foot height from the water surface), such
speeds are quite unlikely. As we believe this was the main barrier in using Horton's equation more widely, we converted the
values of his tables from his lesser-known report (Horton, 1927) into the following expressions for convenience:

$$\Psi(w_0{}^{mps}) = 0.00372w_0{}^3 - 0.0641w_0{}^2 + 0.40396w_0 + 1 \qquad (2d)$$

$$\Psi(w_0{}^{mph}) = 0.00033w_0{}^3 - 0.01281w_0{}^2 + 0.18059w_0 + 1 \qquad (2e)$$

We also converted another table he provided in a much later work (Horton, 1943b) where the values for $\Psi$ varied slightly:

$$\Psi(w^{mph}) = 0.00027w^3 - 0.01162w^2 + 0.17493w + 1 \qquad (2f)$$

The table values for $\Psi$ might possibly be an error in Horton (1943b), but it seems worth pointing out the difference,

however slight. To develop Eqns. 2d-f, we first extracted the values from Horton's table using online scanning software
(https://extracttable.com/), then we fitted it as a two-parameter function with 6 unknowns (see Supplement). We assessed
several methods to develop parametric equations from Horton's tables, such as monkey saddle, logarithmic, and power law
relationships, shifted divergence, rooting behaviours, etc., and were able to obtain a coefficient of determination of 0.99.
However, the functions that provided this fit did not capture the high velocity variations satisfactorily. We were fortunate to

obtain an improved solution with the assistance of Dr. Mikuszeit through Stack Overflow (see Vimal and Mikuszeit, 2021).
The coefficient of determination ($R^2$) of the best formulation was 0.999. Wind velocity, $w_0$, is given by



$$w_0 = f(H, w_H) \tag{3a}$$

$$w_0 = 14.555\, w_H^{1.617}(0.05 + (H - 16.614 w_H + 68.614)^{-0.65})) \tag{3b}$$

where $w_H$ is the wind velocity, as measured by an anemometer at some height H above the ground or above the water surface.

The equation holds for values of height of wind measurement and velocities, $5 \leq H \leq 200$ feet and $1 \leq w_H \leq 30$ miles per hour respectively. These values do not exceed typical conditions. To calculate wind measurements at heights other than $w_0$, since algebraic manipulations cannot be easily used on Eqn. (3b), a bisection search method was used to calculate wind velocities at various heights. This approach was later used for the other equations that relied on wind measurements at different heights. The bisection method converges to within two decimal places with 10 iterations and takes a fraction of a second, so

it can be adopted for simulations over long time periods and over large domains with many grid cells.

### 3.3. Area factor for pan evaporation depending on turbulence and humidity

While using pan evaporation to calculate lake evaporation, an area factor is required. The area factor, F, for pan evaporation uses the concept of *evaporative capacity* ($E_{cw}$) w.r.t. water (note that evaporative capacity in Eqn. 1b is the same but w.r.t. air temperature, and $E_{cw}$ is the same as $E_P$ given in Eqn.1a). It is obtained as the ratio of evaporation from lake $E_L$

to the evaporative capacity ($E_{cw}$):

$$F = \frac{E_L}{E_{cw}} = \frac{C[\Psi V_w - V_b]}{C[\Psi V_w - v_a]} \tag{4a}$$

When the water and air temperatures are identical (this would apply more for small lakes, where the temporal lag in water temperature is negligible), then, $V_w = V_b$ and $v_a = hV_w$, where $h$ is relative humidity given by $v_a/V_a$.

If air and water temperature are equal, then the correction factor F reduces to


$$F = \frac{\Psi - 1}{\Psi - h} \tag{4b}$$

Horton (1943b) deduced that when air and water temperature are not equal, the area correction factor to be related to two ratios ($r$ and $h'$) is similar to relative humidity, where $r = \frac{V_b}{V_w}$, and $h' = \frac{v_a}{V_w}$, as

$$F = \frac{\Psi - r}{\Psi - h'} \tag{4c}$$

These relationships are provided in Horton (1927, p. 162). The influence of turbulence on F is discussed in Horton

(1943). If $p$ is the fraction of time during which turbulent flow prevails up to some considerable height above the ground, under turbulent conditions, correction factor F is given by

$$F = (1 - p) + p\frac{[\Psi - 1]}{[\Psi - h]} \tag{4d}$$





The derivation of Eqn. (4d) is not shown step-by-step in Horton (1943), but it appears that it follows directly from the following equation (Eqn. 5) presented in Sec. 3.4, as indicated by the following quote from Horton:


"[The author] deduced a rational expression for area-factor based on the assumption that near the windward edge of a broad water-surface an unknown fraction m of the emitted vapor is carried to leeward [...]" - Horton (1943)

Contemporary atmospheric boundary layer (ABL) theory helps approximate $p$, which can be determined to a fair degree of accuracy by estimating diurnal variations of boundary layer height (see Stull, 1988).

### 3.4. Vapor blanket characteristics

The vapor blanket is conceptually similar to a viscous sub-layer in open channel flow and is formed due to the existence of a laminar flow layer which horizontally transports moisture in the downwind direction, which leads to its growth in height. The horizontal variation of vapor blanket height, which is in the order of a few meters, is critical when estimating pan evaporation. Pans have a poorly formed vapor blanket because of their small size, as even weak winds can remove the

laminar layer before it is fully formed. Once pan evaporation is corrected for the formation and disturbance of the vapor blanket layer, their use for lake evaporation can be readily justified (Horton, 1927). In the case of both pans and lakes, the vapor blanket characteristics are the same (both are governed by meteorologic factors), but over pans the variation of evaporation over the variable thickness of vapor blanket is more important, while over large lakes, for most of the area concerned, evaporation rate is constant (except in cases of very high humidity and temperature gradients). So the impact of

variable vapor blanket thickness, though present, can be ignored as negligible. It is important to account for the effect of vapor blanket during both daytime (when it's slightly larger) and night-time conditions (see example problem in Horton, 1917a).

**Horizontal variation of vapor blanket**: Understanding the process of vapor blanket formation and accurately quantifying its development and disturbance from the windward fringe of the lake to the leeward side can be considered as one of the main theoretical breakthroughs in Horton's evaporation work. Horton derived an expression (see Eq. 5 below) to

capture where, when and how much the evaporation rate varies across the lake (or pan) surface. Assuming a strip of unit width, the horizontal distance of the vapor blanket before its thickness becomes constant is given by

$$x_c = \frac{1}{mC} log_e \frac{\psi V - v_0}{\psi V - v_c} = \frac{1}{mC} log_e \frac{E_0}{E_c} \tag{5}$$

where $x_c$ is the distance from the windward edge of the water surface where the vapor blanket thickness becomes constant. The horizontal scale of $x_c$ is typically in the order of a few yards. Our calculations show that it can be in the order of a few

meters. $v_0$: vapor pressure at the shore on the windward side; $v_c$: vapor pressure at a distance $x$ downwind; $E_0$: evaporation at the windward shore of the lake; $E_c$: evaporation at $x$; $m$: the fraction of moisture carried by wind action from the shore towards the leeward side of the lake, where vapor blanket thickness quickly approaches a constant value. Typical values of $m$ are given





as:  0: water surfaces broken by waves and over rough land surfaces; 0.3-0.4: gusty winds; 0.6-0.7: steady winds; 1: perfectly horizontal uniform wind (Horton, 1917a).

Though Horton does not provide the steps to derive Eqn. (5), derivations for analogous problems which resemble this equation solved by Horton and others may provide some insight. For convenience of reference, one such derivation by Horton (1927, p. 63) and how it can be interpreted for the derivation of Eqn. (5) is given in the Supplement. Some examples of viscous sub-layer problems in open channel flow are given in Horton et al (1936).

Another useful formula Horton provides is one for calculating evaporation ($E_x$) at any point $x$ along the lake or pan.

Assuming a strip with unit width and length ($x$) downwind along the direction of mean wind, evaporation at the point $x$ is

$$E_x = E_0 e^{-mCx} \tag{6}$$

Average evaporation ($E_{av}$) over the strip from shoreline to the location $x$ over the developing vapor blanket is then

$$E_{av} = \frac{E_0}{mCx}(1 - e^{-mCx}) \tag{7}$$

**Vapor blanket height:** In most cases, vapor blanket thickness is only a few mm, and it is related to wind velocity.

Horton (1943b) presents an equation given by G. I. Taylor (1918). Though Horton's reference has the same title as that provided in reference, the year specified by him (1934) could have been a typo, and the correct reference is likely to be the one given here. After inspecting Taylor's papers from 1934 and conducting a cursory search of his bibliography for similar titles, we did not find the equation Horton provided. From Horton (1943b), vapor blanket thickness ($T_g$, in feet) given by Taylor is apparently $T_g = 0.0293w$, where $w$ is the wind speed at a height of 1 foot in miles per hour.

Horton is among the few hydrologists to rigorously examine the role of the vapor blanket in lake evaporation. So, to conclude this section, a brief synopsis of some of the other studies conducted by other investigators may aid the readers in pursuing further research in this direction. Horton's source for the idea of vapor blanket and its contributions to evaporation rates could perhaps be the Slovenian scientist Josef Stefan (1882):

> "The fact that the amount of evaporation from a basin is proportional not to the surface content but rather to the square root of this surface content leads to the result that evaporation from large water basins is proportionally smaller compared to the evaporation from a small basin. Let us also add that this is true not only for diffusion-driven evaporation but also for convection-driven evaporation. *When an air current moves across a water surface, it will initially lift up large amounts of water vapor as soon as it crosses the boundary of the basin, but then it will not cause much evaporation as it progresses.*"
> - Stefan 1882, p560, emphasis added (own translation)

A derivation similar to that of Eqn. (5) is provided in an analogous problem of diffusion and evaporation by Stefan (1882) who may have inspired Horton's derivation. Stefan, in turn, relates the derivation to two other analogous problems in

heat conduction and electricity. These analogous problems give the germ of the solution for Eqn. (5). Mitrovic (2012) translated an important work conducted by Stefan related to diffusion that has been long forgotten.



The characteristics of the vapor blanket have been studied in only a few other works to our knowledge. Sutton (1934) and Vercauteeren (2011) have considered the shape of the vapor blanket in the windward edge, but its properties with respect to evaporation (and with regards to turbulence, convection, etc.) over lakes were not unexplored. Millar's (1937) apparently rigorous study of the vapor blanket was not accessible to us (we were unable to obtain a copy of the paper), but a summary is provided in a USGS report (1954; see chapter on Mass Transfer Studies by Marciano and Harbeck) which shows Millar's equations. They indeed seem to resemble Stefan's work on diffusion. Finally, there is an indirect reference to vapor blanket in Peter Eagleson's textbook on *Dynamic Hydrology* which supposedly includes a description of the vapor blanket as a conceptual thin layer, and it is described with a schematic, but no sources were given (Eagleson, 1990, Fig. 12-1, p. 213).

### 3.5. Separable physical factors in the evaporation equation

**Role of pressure (evaporation change with altitude):** To understand the role of vapor removal and diffusion, for convenience we can consider a general form of equations (1) and (2). Ignoring convection, inserting $\Psi$ from Eqn. (2a), into Eqns. (1a) and ignoring the suffixes of water and air for simplicity, the general evaporation equation is given by

$$E = C\left[(\mathcal{H} - e^{-kw_0})V - v\right] \tag{8a}$$

If $\mathcal{H}$, as given by Horton (drawn from Dalton), can be taken as a constant 2, then Eqn. (8a) can be factored into

$$E = \underbrace{C(V - v)}_{Diffusion} + \underbrace{C(1 - e^{-kw_0})V}_{Vapor\ removal} \tag{8b}$$

By separating Eqn. (8a) into its physically meaningful parts as shown in Eqn. (8b), one can account for the role of barometric pressure which impacts only one of the terms (free diffusion, which is the first part here). When pressure changes with altitude, the first term here is adjusted for pressure drop which solely impacts free diffusion. Horton's rationale is as follows:

> "It is evident that in order to determine the effect of change in barometric pressure on evaporation, other things equal, its effect on vapor removal by diffusion, which is always present, and its effect on vapor removal by wind-action, must be considered separately. This may readily be accomplished by the use of an evaporation formula published some years ago" – Horton (1934)

An inverse relationship between diffusion and pressure was first proposed by Thomas Tate (1862) and later derived by Stefan (1881). If $B_0$ and $B$ are barometric pressures at datum (sea level) and pressure at a given elevation respectively, the evaporation equation, according to Horton (1934), becomes

$$E = C\left(\frac{B_0}{B}\right)(V - v) + C(1 - e^{-kw_0})V \tag{8c}$$




The second part represents enhanced vapor emission facilitated by vapor removal from the vapor blanket, which can be by wind or convection: wind's influence is independent of barometric pressure (Horton, 1934). The relationship between convection and barometric pressure was not known to him, and he had an argument to not investigate further:

> "The relation of barometric pressure to convective vapor removal has apparently not been
> studied. Since convection is, in general, not present when there is strong wind-action, it
> will not be considered here." – Horton (1934)

    Under humid conditions, Horton (1934) suggested that the role of wind-induced vapor removal may be several times higher than that of still air, but it does not appear that this effect is explicitly accounted for in his equation.

### 3.6. Experimental precision

The precision that went into Horton's experimental measurements is quite remarkable. He performed detailed experiments on the melting of snow considering dozens of physical variables measured at 10-20-minute intervals (Horton, 1915). These experiments, together with his earlier study on evaporation from snow (see Horton, 1914) seem to have contributed to his later experiments on condensation (see Horton, 1917a). He developed instruments to measure minimum and maximum daily temperatures of water surface and a geometrical approach for snow temperature (Horton, 1919b). To 625 cross-check his daily snow measurements, he made additional measurements at an accuracy of $1/5^{th}$ of a degree at hourly intervals to cross check the diurnal (min and max) daily snow temperature readings (Horton and Leach, 1934). He used graphical methods to calculate vapor pressure and humidity which give values to within 1-2% accuracy (Horton, 1921). Some evaporation measurements to cross-check his evaporation calculation (see Horton, 1927, pp. 150-155) were made to ~$1/1000^{th}$ of an inch precision.

## 4.  Evaluation of Horton's evaporation method

### 4.1.  Evaluating on an Arctic lake with observed and disaggregated vapor pressure

    High latitude lakes are quite important in the context of accelerated Arctic warming (Smith et al, 2005), as the region is besprinkled with numerous tiny lakes, where the mean evaporation for each lake may vary appreciably due to the variability of vapor blanket thickness (Eqns. 5-8), which means that the role of the vapor blanket cannot be ignored. In the domain of 635 Canada and Alaska alone, there are over 13 million lakes measured at Landsat resolution (approximately ~0.1 hectares, but varies by latitude), and perhaps many more at finer scales. Horton (1934) noted that high latitude evaporation processes may be quite different from mid latitudes, because available water at the surface may be altered by condensation processes, and the predominant evaporation surface is snow, especially above the snow line (Horton, 1934). So it follows that the methods of midlatitudes cannot be directly applied, though Horton believed that his evaporation method is generalizable for sub-zero 640 conditions and condensation (unlike the other empirical equations for evaporation).



We tested Horton's evaporation equation on Baker Creek in subarctic Canada where 30-minute meteorological data were available as measured over the lake as well as near the lake (see Spence and Hedstrom, 2018 for data description and measurement heights). For vapor pressures of air measured in either location, the difference in evaporation was slight. To evaluate the performance of Horton's equation, following Singh and Xu (1997), we selected five other equations that resemble

Horton's equation and calibrated them by treating all the coefficients as free parameters, preserving only the shape of the equation. We selected 5 equations (see Table 3 shown later) with various shapes: Konstantinov, Dalton, Meyer, Rohwer, Penman. Note that Penman referred here is not the combined equation, but only a part of the combined equation (aerodynamic) provided in Penman's original work (Penman, 1948). Most empirical Dalton-type formulas do not include a temperature deficit term, except few that are of the type of Konstantinov (1968). The general forms of the equations are given in Table 3.

Actual vapor pressure of the air is one of the most important variables which is difficult to obtain. So, to understand the robustness of the various methods to errors in this variable, in addition to using observed measurements available for the test site, we calculated actual vapor pressure as a function of solar geometry, diurnal temperature range and seasonal precipitation (see Bennet et al, 2020 and Bohn et al, 2013). The data for this were drawn from our previous work (Vimal et al, 2019).

We used a bootstrap approach to get the mean (μ) and standard deviation (σ) for coefficient of determination ($R^2$) and percentage bias (% of mean absolute percentage error), where we sampled 50%, 75% and 100% of the record length and 50 random samples with replacement for each length, and 11 time scales (30 minutes to 2 months), in total 1650 random bootstrap samples. For all these combinations, the time period of analysis was 8 April, 2009 to 20 September, 2016. Missing values were ignored, and data coverage mostly represents Summer months (further details are in Spence and Hedstrom, 2018).

**Table 1: Performance metrics ($R^2$ and % bias) for evaporation methods using observed data inputs. Darker shades of teal and pink highlight the good results.**

| Method | Metric | 30min μ | 30min σ | 1H μ | 1H σ | 4H μ | 4H σ | 12H μ | 12H σ | 1D μ | 1D σ | 3D μ | 3D σ | 1W μ | 1W σ | 3W μ | 3W σ | 1M μ | 1M σ | 2M μ | 2M σ |
|---|---|---|---|---|---|---|---|---|---|---|---|---|---|---|---|---|---|---|---|---|---|
| | Time / N | 14,828 | | 8,096 | | 2,458 | | 1,004 | | 543 | | 202 | | 91 | | 34 | | 24 | | 15 | |
| Horton | $R^2$ | 0.51 | 0.01 | 0.53 | 0.02 | 0.61 | 0.02 | 0.77 | 0.02 | 0.85 | 0.03 | 0.87 | 0.04 | 0.81 | 0.05 | 0.85 | 0.04 | 0.86 | 0.04 | 0.91 | 0.02 |
| | % Bias | 22.58 | 0.26 | 22.27 | 0.30 | 21.13 | 0.55 | 17.40 | 0.93 | 15.51 | 1.54 | 18.47 | 3.31 | 25.75 | 2.87 | 22.39 | 4.74 | 27.64 | 4.81 | 26.89 | 6.59 |
| Konstantinov | $R^2$ | -0.12 | 0.02 | -0.12 | 0.03 | 0.06 | 0.04 | 0.45 | 0.04 | 0.69 | 0.03 | 0.77 | 0.04 | 0.74 | 0.04 | 0.79 | 0.04 | 0.82 | 0.03 | 0.87 | 0.03 |
| | % Bias | 36.66 | 0.22 | 36.43 | 0.30 | 33.61 | 0.52 | 28.99 | 0.85 | 23.21 | 0.96 | 22.36 | 1.85 | 26.66 | 1.96 | 21.63 | 3.35 | 27.73 | 3.38 | 27.73 | 5.83 |
| Dalton | $R^2$ | -0.10 | 0.02 | -0.09 | 0.03 | 0.07 | 0.04 | 0.44 | 0.04 | 0.65 | 0.04 | 0.77 | 0.06 | 0.67 | 0.07 | 0.75 | 0.06 | 0.80 | 0.04 | 0.87 | 0.02 |
| | % Bias | 35.62 | 0.26 | 35.39 | 0.34 | 32.92 | 0.63 | 27.84 | 0.92 | 23.10 | 1.63 | 24.29 | 2.42 | 31.82 | 2.81 | 25.76 | 4.69 | 30.65 | 4.09 | 29.45 | 6.49 |
| Meyer | $R^2$ | -0.05 | 0.02 | -0.05 | 0.03 | 0.12 | 0.04 | 0.49 | 0.04 | 0.73 | 0.03 | 0.79 | 0.04 | 0.74 | 0.05 | 0.78 | 0.04 | 0.81 | 0.04 | 0.88 | 0.02 |
| | % Bias | 35.48 | 0.22 | 35.18 | 0.29 | 32.42 | 0.52 | 26.98 | 0.87 | 21.74 | 1.19 | 24.72 | 2.28 | 29.54 | 2.39 | 25.91 | 4.28 | 30.42 | 4.13 | 28.47 | 6.16 |
| Rohwer | $R^2$ | -0.06 | 0.02 | -0.05 | 0.03 | 0.12 | 0.03 | 0.49 | 0.04 | 0.73 | 0.03 | 0.79 | 0.04 | 0.74 | 0.05 | 0.79 | 0.04 | 0.81 | 0.04 | 0.88 | 0.02 |
| | % Bias | 35.53 | 0.21 | 35.25 | 0.29 | 32.47 | 0.50 | 26.98 | 0.84 | 21.68 | 1.12 | 24.83 | 2.28 | 29.53 | 2.37 | 25.80 | 4.27 | 30.36 | 4.10 | 28.38 | 6.16 |
| Penman | $R^2$ | -0.05 | 0.02 | -0.05 | 0.03 | 0.12 | 0.03 | 0.49 | 0.04 | 0.73 | 0.03 | 0.79 | 0.04 | 0.74 | 0.05 | 0.78 | 0.04 | 0.81 | 0.04 | 0.88 | 0.02 |
| | % Bias | 35.47 | 0.21 | 35.16 | 0.29 | 32.39 | 0.51 | 26.93 | 0.85 | 21.67 | 1.13 | 24.75 | 2.28 | 29.54 | 2.39 | 25.92 | 4.28 | 30.39 | 4.10 | 28.40 | 6.15 |

Table 1 shows that Horton's method is substantially more accurate than the other five methods of varying complexity consistently across timescales and sample sizes.





665        **Table 2: Performance metrics ($R^2$ and % bias) for evaporation methods using reanalysis-based disaggregated actual vapor pressure. Darker shades of teal and pink highlight good results.**

| Method | Time | 1D | | 3D | | 1W | | 3W | | 1M | | 2M | |
|---|---|---|---|---|---|---|---|---|---|---|---|---|---|
| | N | 543 | | 202 | | 91 | | 34 | | 24 | | 15 | |
| | Metric | μ | σ | μ | σ | μ | σ | μ | σ | μ | σ | μ | σ |
| Horton | $R^2$ | 0.82 | 0.03 | 0.86 | 0.04 | 0.81 | 0.05 | 0.84 | 0.04 | 0.86 | 0.04 | 0.91 | 0.04 |
| | % Bias | 18.93 | 1.39 | 19.33 | 2.43 | 25.85 | 2.90 | 21.72 | 4.09 | 19.23 | 4.19 | 16.22 | 3.84 |
| Konstantinov | $R^2$ | 0.62 | 0.04 | 0.68 | 0.04 | 0.68 | 0.05 | 0.71 | 0.06 | 0.70 | 0.07 | 0.73 | 0.11 |
| | % Bias | 30.90 | 1.01 | 32.16 | 1.70 | 32.29 | 2.58 | 25.84 | 3.60 | 28.09 | 4.18 | 23.77 | 3.19 |
| Dalton | $R^2$ | 0.55 | 0.05 | 0.65 | 0.05 | 0.61 | 0.08 | 0.64 | 0.07 | 0.67 | 0.07 | 0.69 | 0.12 |
| | % Bias | 32.60 | 1.33 | 32.29 | 2.12 | 34.77 | 3.14 | 29.04 | 4.10 | 28.08 | 3.94 | 25.28 | 2.98 |
| Meyer | $R^2$ | 0.64 | 0.04 | 0.68 | 0.04 | 0.69 | 0.05 | 0.72 | 0.06 | 0.69 | 0.07 | 0.72 | 0.11 |
| | % Bias | 30.53 | 1.13 | 31.74 | 1.88 | 32.66 | 2.63 | 27.22 | 3.93 | 27.73 | 4.20 | 23.71 | 2.91 |
| Rohwer | $R^2$ | 0.64 | 0.04 | 0.69 | 0.04 | 0.69 | 0.05 | 0.72 | 0.05 | 0.70 | 0.07 | 0.72 | 0.11 |
| | % Bias | 30.33 | 1.13 | 31.61 | 1.87 | 32.50 | 2.65 | 27.08 | 3.93 | 27.60 | 4.21 | 23.51 | 2.92 |
| Penman | $R^2$ | 0.64 | 0.04 | 0.68 | 0.04 | 0.69 | 0.05 | 0.72 | 0.06 | 0.70 | 0.07 | 0.72 | 0.11 |
| | % Bias | 30.53 | 1.13 | 31.74 | 1.88 | 32.66 | 2.64 | 27.23 | 3.93 | 27.70 | 4.21 | 23.67 | 2.90 |

Surprisingly, Horton's method outperforms other methods even when using estimated input vapor pressure (Table 2) while the other 5 methods use local measurements (Table 1). It must be noted that previous studies have shown that vapor

670        pressure near water bodies (e.g. coastal regions) has a large bias and uncertainty (see Bohn et al, 2013), which makes the result even more surprising. A reason for the poorer performance of other methods could be that we estimated wind velocity at various heights by back-calculating using Eqn. (3b) and the bisection method previously mentioned (Sec. 3.2). Another reason could be the dependence of vapor pressure measurement on observation height for some, even if not all, of the other methods. Konstantinov's equation depends on wind speed at ground height (same as Horton's method), and uses more input variables

675        related to temperature, and yet does not perform better. We do not draw bald conclusions directly from Tables 1 & 2, before testing under multiple catchments and lakes of wide-ranging meteorological conditions. However, if this result holds across various locations and regions, as we will show in Sect. 4.5 more generally, then we can arrive at a few conclusions: 1) that Horton's formula is robust against over-fitting of errors making it more physically based; and 2) the variable vapor pressure deficit (VVPD) term, unique to Horton's evaporation formula, is a better control on evaporation than VPD.

**4.2. Generality of the method**

We use the term generality in the following connotations: 1) Parameter certainty: i.e. how relatively unchanging the parameters in the calibrated equations are across wide ranging conditions, time averages (mean of evaporation is considered when time averaging, so effect of time in parameters is ignore), and record lengths; 2) how well it performs in wide ranging conditions across various meteorological conditions and altitudes; 3) How well it performs across continental which follows from both (1) and (2). The ability for a method to generalize across such conditions shows that the method is not an empirical

fit, but has a rational or physical basis.





### 4.3. Parameter certainty



If the parameter values are unchanging or have only a slight variability, they can be assumed to possess a physical meaning which does not need site-specific tuning (or calibration). Such unchanging values are termed constants and identifying such constants is common in Physics. Of the three connotations of generality we are interested in, parameter certainty is the most important. In all the six methods we compared, there were 17 parameters, and each one was tuned for each of the 1,650 bootstrap samples using a vectorized approach (see Sec. 4.1 for breakdown of sample size and record lengths). In total, this represents 1,68,300 tuned parameters which are summarized in Table 3 (shown below). To make their comparison straight forward, the time unit of reference observation was kept identical to the native resolution, e.g. daily or monthly evaporation values were averaged into units of mm per 30 minutes, which allows us to compare values of parameters across methods and time scales. Some outliers in parameter values were found (possibly due to errors in data) but were removed using the same criteria ($10^{th}$ percentile) for all 6 methods each considered independently. The last column here shows normalized values of variability ($\sigma/\mu$) as a percentage, which can be compared across methods.

**Table 3: Parameter uncertainty comparison between six evaporation formulas (mean $\mu$ and $\sigma/\mu$)**

| Evaporation | Equation | Parameter | Mean ($\mu$) | $\sigma/\mu$ (%) |
|---|---|---|---|---|
| Horton | $C[(\mathcal{H} - e^{-kw_0})V - v]$ | H | 1.71 | 1.3% |
| | | K | 0.13 | 4.3% |
| | | C | 0.18 | 12.9% |
| Meyer | $C\,(V-v)(A - u_9/B)$ <br> $u_9: wind\ at\ 9m\ height$ | C | -0.06 | 1.6% |
| | | A | -2.39 | 4.3% |
| | | B | 4.70 | 7.2% |
| Penman | $A(V-v)(B + Cu_2)$ <br> $u_2: wind\ at\ 2m\ height$ | A | 0.16 | 10.5% |
| | | B | -0.33 | 5.1% |
| | | C | -0.06 | 5.6% |
| Rohwer | $A\,(B - CPa)(D + E*u_0)(V-v)$ <br> Pa: pressure <br> $u_0: ground\ wind$ | A | 1.03 | 3.8% |
| | | B | 1.03 | 20.3% |
| | | C | 0.98 | 20.3% |
| | | D | 0.56 | 6.1% |
| | | E | 0.92 | 5.1% |
| Konstantinov | $\left[A\,\dfrac{(\theta - \theta_a)}{u_0} + Bu_0\right] * (V-v)$ | A | 0.09 | 20.2% |
| | | B | 0.03 | 9.8% |
| Dalton | $C\,(V-v)$ | C | 0.39 | 18.4% |


Among all parameters, the parameter $\mathcal{H}$ has the most unchanging value (1.71) and the smallest (1.3%) relative variability ($\boldsymbol{\sigma/\mu}$ **%**), while average of all other parameters is 9.5%, which shows that it is the most generalizable and requires the least site-specific tuning among the 17 parameters considered across all 6 methods. The value for $\mathcal{H}$ that Horton originally prescribed was 2, drawing from Dalton's experiments (see quote in Sec. 2.2.2). The other two parameters of Horton's equation





are not particularly more certain than the parameters of other equations. Meyer's equation, which relies on wind speed at 9m height, has one of the parameters (C) that performs nearly as well as Horton's $\mathcal{H}$, with 1.6% variability, but it has no physical meaning as it has a negative value.

**Previous investigations on $\mathcal{H}$:** To our knowledge, there is no other evaporation formulation that captures the role of $\mathcal{H}$, though aspects of its role have been observed previously. Horton's source for $\mathcal{H}$ could be regarded as Dalton (1802).

Dalton conducted his experiments in a single site in high and low evaporation conditions and high and low temperatures, so our result (1.71) can be said to be more robust than Dalton's, as our bootstrap sampling strategy accounts for more wide-ranging conditions. Even so, the value of 1.71 may need confirmation from several lakes across latitudes to ascertain its value. This value, interestingly, agrees very closely with Schübler's (1831) experimental observations that evaporation accentuated by wind during Summer was 1.7 times greater (Brutsaert, 1982). The parameter $\mathcal{H}$ appears to be a significant development in

lake evaporation physics, and can be designated as *Horton constant,* sharing credit with Dalton and Schübler.

### 4.4. Estimates across altitudes and sub-zero temperature conditions

Horton claimed that his method was rational (physical) in that it is robust to conditions outside for which it was used (Horton, 1927; p159), which the other empirical methods of his time were not (e.g. that by Carpenter and Fitzgerald, see Fitzgerald, 1886), as most were tuned for local conditions. He investigated the role of condensation rates, evaporation from

snow surfaces (Horton, 1914), temperature deficits and wind speed in high altitude and polar regions (Horton 1934). In a Snow Conference paper (Horton, 1943) he comments on the processes involved in evaporation from snow that includes independent variables that depend on latitude and altitude, which were not known with certainty. When lake surfaces are partially covered with ice, he recommends using a weighted average of lake water and ice temperatures, for partially frozen lakes. The role of thickness of ice on air-water temperature relationship was observed, i.e. thicker ice brings air and ice

temperature closer. Additional factors that influence evaporation under such conditions could be the percentage, intensity, and duration of laminar and turbulent air flow, which depend on latitude and elevation (*ibid*), and also other physical factors due to snow and ice, that is A) area exposed to air (vs projected area from snow surface) due to influence of snow porosity may increase evaporation; and B) the disproportionate departure of air temperature much above ice temperatures compared to water temperature. Horton suggested that these additional factors may require a separate treatment (Horton, 1934).

### 730 4.5. Evaluating Horton's evaporation results over Continental U.S.

Horton used 112 pan evaporimeters' data over the continental US and plotted precipitation, evaporation and runoff into one figure sliced by longitudes (see figure in Horton, 1943). We re-plotted his chart together with a land surface model results simulated at over 200,000 model grid locations over the continental US by Livneh et al (2013). We aggregated the model results the same way as Horton did by 2-degree grid boxes. Surprisingly, the curves for P, E, and Q are remarkably





similar (see Fig. 2). We further aggregated data into three climate normals, i.e. three 30-year averages from 1921 to 2010 to see whether there exist long term climate change influences, but found none - this could possibly be an inherent issue with the Livneh et al, 2013 dataset, which possibly is de-trended. The record lengths of Horton's data were variable, so they are not shown, but they are in the order of magnitude to be regarded as climate normals, i.e. long-term climate average.

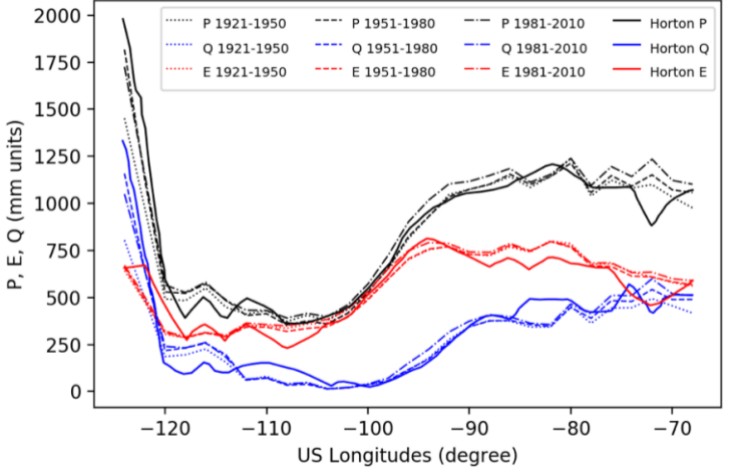

**Figure 2: Comparison of con-US 2-degree average values of precipitation (P), Evapotranspiration (E), and runoff (Q) estimates: replotting the chart from Horton (1943b) together with Livneh et al (2013) over three climate normals.**

The difference in evaporation is substantial in the Great Lakes region (between longitudes -90 and -80), though precipitation seems to be similar, and this may be explained as follows: Cleveland and Chicago, which are on different sides of the Great Lakes, may have a similar temperature (except in Winter), but the number of sunshine hours (and cloud cover)
may change significantly between the two places (see Jenson and Haise, 1963). Some of these factors were directly and indirectly accounted for in Horton's estimation of evaporation from the Great Lakes. For example, he considered wind data from multiple locations and performed some interpolation-based corrections. Also, the land surface model results were masked out for the Great Lakes pixels, so it is possible that the evapotranspiration of that longitude band on average is greater than the evaporation from Great Lakes which may explain the difference. Larger lakes, as noted by Stefan (1882) and suggested
by Horton's formula (Eqn. 5), may possibly yield a lower total evaporation than the rest of the land surface, which is quite unintuitive, but for the scale of the Great Lakes, this cannot be ascertained as there may be numerous other factors that come into play. However, we can conclude that the evaporation formula does generalize over continental scales owing to the remarkable similarity seen in Fig. 2.





## 5. Discussion

### 5.1. Horton's contribution to lake evaporation physics

While this paper highlights a century-old method, we do not fail to recognize that advancements in evaporation theories of the last century have been stellar: one needs to only look at the number of numbers (mostly dimensionless) that are used to represent the physics that control evaporation - Dalton, Reynold, Prandtl, Taylor, Karman, Stanton, Schmidt, (Flux) Richardson, Peclet, Nusselt, Sherwood, Raleigh to name some (see Pasquill, 1942; Hess, 1979, and Brutsaert, 1982 for an introduction to many of these developments). Besides the fields of aero-, thermo- and hydrodynamics where most of these numbers emerged, there have been also great strides forward in the kinetic theories of evaporation (see Gerasimov and Yurin, 2018). One can argue that progress would lead to unification of these numbers into a smaller set. Nevertheless, in the quest for the smaller set, among the candidate numbers, we believe two of Horton's core contributions discussed in this paper could be considered for their fundamental relevance to lake evaporation estimation: 1) the ratio $E_0/E_c$ in Eqn. (5) which represents the ratio of evaporation at the fringe of the lake to evaporation where the vapor blanket acquires a constant thickness; 2) $\mathcal{H}$, the seemingly constant coefficient (see Table 3), the value of which was prescribed by Horton as 2 (or a little lower as we find, 1.7), which is arguably what makes the VVPD term a better independent control on evaporation than VPD. These two, we suggest, could be called the *Horton ratio* and the *Horton constant for lake evaporation*, respectively. The former could also be credited to Stefan (1881); see also Mitrovic (2012) who re-discovered another century-old problem credited to Stefan, and the latter to John Dalton (1802) and Gustav Schübler (1831). It appears that Horton provided the first quantitative treatment highlighting the importance of these two values for lakes.

### 5.2. Can Horton's evaporation formula replace other methods?

Among the 5 equations we evaluated, Meyer's, Rohwer's and Penman's equation shapes and results differ but slightly. Expectedly, Konstantinov's (1963) method, which draws additional information from a temperature deficit term, in addition to VPD and wind (as done in other methods), has the second highest complexity and performs the second best, while Dalton's method (the simplest one) is the poorest. What we have shown here suggests that Horton's equation can indeed replace these other methods. A question that begs to be answered here is: should Horton's evaporation equation for lakes be preferred over the Penman (combination) equation, especially in the context of continental scale land surface modeling? Before answering this question, it is worth noting that Penman's formula is indeed adapted from Rohwer's (1931) formula, who in turn in his work commented on Horton's evaporation formula, saying that,

> "From a theoretical standpoint [Horton's] formula is worthy of consideration, but, as the values of the constants in the formula have not been definitely determined, the practical value of the formula is small"



Our answer to this question from this study is that it could be for the following reasons. **1) Horton's VVPD can replace VPD:** the aerodynamic part of the Penman equation invariably depends on the VPD term which, as we showed in Sec. 4.1, will indeed be less accurate than Horton's VVPD. **2) Separability of barometric pressure:** the Penman aerodynamic component is weighted by psychrometric constant (essentially a barometric pressure term), which plays a role in diffusion but not aerodynamic action. As shown in Eqn. 8c, an inverse barometric pressure term may be added to only the diffusion term, which is separable from wind action, which is possible with Horton's equation but not the other aerodynamic formulas hitherto used in various combination methods (Penman and others). **3) Error in energy variables:** the energy balance approach relies on variables such as surface radiation and ground heat flux (which depend on cloud cover, ground heat exchange, etc.), which are prone to errors. Furthermore, there exist first-order issues with energy budgets because of errors in a crucial variable, open water albedo, which varies as a function of sun's angle (see field experiments by Sivkov, 1971). Seasonal variability can be up to a factor of 7, but most lake schemes do not account for this variability: for example, the lakes energy scheme of Bowling and Lettenmaier (2009) uses a constant albedo value for open water (similarly in Hostetler and Bartlein, 1990; and Croley, 2012). **4) Horton's method depends on water temperature data and not radiation**: using water surface temperature data for evaporation has the crucial advantage that it can be directly measured from space (see Sharma et al., 2015), especially for large water bodies, with a fair degree of accuracy (~1.15 °C for small lakes and 0.45 °C for large lakes). Rapid mixing of surface water due to wind, and vertical density gradients (see experiments on stratification by Gregory, 2012) together favor surface water temperature to equalize quickly across the surface. This is especially true in small lakes where surface temperature can be considered as uniform. Over large lakes, temperature varies with bathymetry due to variable rates of vertical mixing in large lakes - however, this variability only depends on lake bathymetry which can be treated as a static parameter, and heat exchange can be modeled or observed with better accuracy in larger lakes from space observations (as noted before). On the other hand, in a study by Rahaghi et al. (2019), it was shown that radiation at the surface of a large Swiss lake (Lake Geneva) varied on the order greater than 40 Wm$^{-2}$ in different parts of the same lake, which is quite a significant error for a large lake and was attributed to shading effect by clouds, a dynamic error. In terrestrial hydrology, where radiation budget is calculated from temperature (e.g. Bohn et al, 2013), Horton's method has a particular advantage. These arguments make a strong case for favouring Horton's equation over the combination method, for both large and small lakes.

### 5.3. Should we revisit the evaporation paradox?

The relationship between pan and actual evaporation is a topic of great importance today in the wake of accelerated climate warming. There is unanimous consensus that pan evaporation is reducing globally, while in a warming climate the opposite is generally expected, which is known as the *evaporation paradox* (Roderick and Farquhar, 2002). A friendly introduction to the topic is given in Singh (2016, Chapter 42.2.3). This paradox is explained by evaporation observations in larger scales across sites of variable moisture availability, considering how energy is redistributed between latent and sensible



heat based on moisture availability. This paradox is considered resolved by Bouchet's (1963) principle of complementarity, which shows the relationship between pan, actual and theoretical evaporation. Morton (1994), Szilagyi et al. (2017), Brutsaert and Yeh (1970) and Brutsaert (1982, 2015) further extended the work by Bouchet (1962). In studies that involve pans, including several that are related to the evaporation paradox, pan evaporation calculations are often done with a static pan correction parameter, but as Horton shows very clearly, it would be quite wrong to use a static parameter (Horton, 1917a).

The explicit role of vapor blanket has been ignored in these studies except perhaps indirectly (as moisture availability is related to atmospheric humidity, which influences vapor blanket characteristics). A table in Maidment (1992; Table 4.3.1., Chapter 4 on Evaporation by Shuttleworth) taken from Doorenbos and Pruitt (1977) provides a quasi-quantitative guidance on pan correction as a function of humidity values and a scale similar to Beaufort wind force scale (i.e. light, moderate, strong winds). However, Horton's quantitative treatment and physical explanations for the differences in evaporation rates from pan to lake

precedes Doorenbos and Pruitt (1977) by half a century. Furthermore, Horton's insights on vapor blanket's physical properties (Sec. 3.4) and the area factor, F (Sec. 3.3), shed a new light on the evaporation paradox and generalizes it beyond standard pan sizes. Considering these, it seems that a revisit to explain the evaporation paradox is warranted.

## 6. Conclusions

Horton's century-long forgotten works on lake evaporation seem to have great contemporary value for the theoretical

insights they offer and for their relevance in modelling lakes of all sizes. The fine-scale precision afforded by Horton's *"law of the wall"*-type equation (Eqn. 5) and Eqns. 6 & 7 for vapor blanket characteristics credited to Josef Stefan and him appears to be essential to estimate evaporation in small lakes and pans, and using pan evaporation as a proxy for large lakes. From these equations, considering the importance of the *Horton ratio* ($E_0/E_c$), taken together with the area factor F (Eqns. 4a-d) for pan evaporimeter measurements, an opportunity arises to revisit the complementarity relationship between pan and lake

evaporation and the so-called evaporation paradox. More generally, Horton's improved formulation that relies on the Horton Constant $\mathcal{H}$ and VVPD (credited to John Dalton, Gustav Schübler and him), due to the dynamic wind factor $\Psi$ (Eqns. 2 a-c), may partially or fully supplant other evaporation equations that rely on VPD, owing to its better generalizability (local to continental, across time scales and latitudes). We believe that Horton's evaporation method was largely overlooked and forgotten because the tables needed for their proper use were unavailable widely. Therefore, in this paper we present the

parametric forms of his ground wind velocity experimental results (Eqns. 2d-f and Eqn. 3b), which may serve as a ground reference for wider use of his method. Considering all this, our main conclusion is that Horton's (1917a) claim of having developed a superior evaporation method (Eqns. 1a-c) seems to hold even today: we believe that his method, which was heuristically related to physical laws, is an improvement over other known lake evaporation formulae.



## 7. Closing note

As a closing note, to entertain the *History of Hydrology Special issue* readers, we would like to highlight an amusing historical anecdote that came out of Horton's detailed evaporation study. In the early morning, over warm lakes in a cold climate, when the wind is calm and laminar flow of wind on the surface of the lake feeds moisture into convective plumes of vapor, they appear as columns of about 10 inches in diameter and over 4 feet in height. Horton got the rare chance to witness these apparitions in his early morning observations – he calls them the dancing columnar vapor drift (Horton, 1933). He notes

that this phenomenon, as also previously noted by a German scientist (Dr. Johannes Walther), may be a curious explanation of the myth of the Greek deity Venus' origin and that of the dancing Nereids upon Greek waters.

     We hope the various observations and conclusions drawn here to highlight the value of Horton's lake evaporation works will be developed further. We also hope this serves to rekindle the interest of readers to (re-)discover Horton's contributions to lake evaporation in addition to his broader published and unpublished works.


*Data availability.* Data sets used in this article are publicly available (see citations in text), and codes are available upon request. The updated bibliography of Horton is available upon request.

*Competing interests.* The authors declare that they have no conflict of interest.


*Special issue statement.* This article is part of the special issue "History of hydrology" (HESS/HGSS inter-journal SI). It is not associated with a conference.

*Author contributions.* SV conceptualized the study, performed simulations, and prepared the manuscript. VPS conducted

literature search, design of simulations, interpretations of Horton's derivations, and edited the manuscript.

*Acknowledgments.* Acknowledgements are due to: 1) Ms. Anna Bonazzi, from UCLA Germanic Languages, for translation of a German text (Stefan, 1882) cited here, for proof-reading and for sharing some ideas which improved this work*;* 2) Dr. Elizabeth Clark (formerly at UCLA) for preparing an earlier version of Fig. 2, and for sharing her collection of 135 of Horton's

titles; 3) Dr. Eric Sheppard, UCLA, for inspiring the work on scientific knowledge production in a graduate class, and also for providing numerous comments on an broader version of this work; 4) Dr. Nikolai Mikuszeit (see Vimal and Mikuszeit, 2021), who offered his feedback via Stack Overflow and provided a superior solution for Horton's wind velocity height correction; 5) Dr. Keith Beven, Lancaster University, handling Editor, provided numerous comments to improve the manuscript; 6) Dr. Dennis Lettenmaier, UCLA, for sharing some interesting perspectives and articles by Horton.



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
