# Peer review of "Re-discovering Robert E. Horton's Lake Evaporation Formulae: New Directions for Evaporation Physics"

_Hydrology and Earth System Sciences, 2021_

## Referee Comment (RC2)

Review by Femke Jansen

The manuscript by Vimal and Singh brings back to our attention the century old Horton's lake evaporation formula. The authors give a thorough historical overview on how the formula was developed and how it relates to other evaporation methods of varying complexity. The authors show us that Horton's formula outperforms the other methods.

I have appreciated reading the manuscript that has been written in a story-telling form including quotes of the original papers of Horton. This provides the reader a good overview and sense on how the authors have reconstructed how Horton's formula was developed and subsequently fell into oblivion. The authors managed to re-awaken the use of it by applying it on data from a subarctic Canadian catchment and found that the use of the variable vapor pressure deficit (VVPD) term introduced by Horton is of added value compared to the use of only VPD which is frequently used in other evaporation methods.

In short, I have read the manuscript with great interest and I think it fits the special issue *History of hydrology.* My suggestion is to publish the manuscript with very minor revisions for which I provide feedback in my comments below.

General comments
-   Please, provide units when explaining the variables of equations for clarity. In some cases it is given (e.g. p.12 L.358), but in most not.
-   The order of the tables as they are mentioned in the text is the other way around of the appearance of the tables itself.

Specific comments
-   Is there a specific reason why the authors are using $\theta$ for temperature, instead of the commonly used $T$? To my knowledge $\theta$ is more commonly used to indicate potential temperature.
-   p.11 L.315 and L.325; w.r.t. – don't write as abbreviation
-   p.14 L.422-425; in more recent past, there are many other studies that have found Dalton's method to work well. Especially in the oceanographic community it is widely used. The authors could refer to that as well for a bit of nuance.
-   P.16 L.490; the reference of Vimal and Mikuszeit, 2021, is not included in the reference list of the manuscript.
-   P.18 L.534; humidity and temperature gradients is probably referring to *horizontal* gradients.
-   P.18 L.533/534; do the authors have a reference that underpins the statement that evaporation rates are constant over large lakes?
-   P.18 L.534-539; First, the authors state that horizontal variability of the thickness of the vapor blanket is negligible, while the next paragraph is dedicated to the importance of horizontal variation and it is mentioned that this is the main theoretical breakthrough of Horton. Please, make this transition more clear or explain better.
-   P.20 L.583; typo: Vercauteeren --> Vercauteren
-   P.20 L.584; do the authors mean 'were not explored' instead of 'were not unexplored'?

---

## Author Comment (AC1)

**Author Responses**

**Re-discovering Robert E. Horton's Lake Evaporation Formulae: New Directions for Evaporation Physics**

Solomon Vimal[1], Vijay P. Singh[2]

[1]Department of Geography, University of California, Los Angeles, CA, 90049, USA

[2]Department of Biological and Agricultural Engineering & Zachry Department of Civil and Environmental Engineering, Texas A&M University, College Station, Texas 77802-2117, USA

Correspondence to: Solomon Vimal (solomonvimal@ucla.edu)

Dear Dr. McMahon,

Thank you for your valuable review comments. Prof. Singh and I reviewed your comments, and we provide our point-by-point response below.

For convenience of reading, we have indented your comments, reduced font size, italicized, and changed color to blue.

Thanks and best wishes,

Solomon Vimal and Vijay P. Singh

**Responses to Reviewer 1 Comments**

> *TM: I am privileged to review this excellent article. The authors have provided an eclectic assessment of Robert E Horton's lake evaporation formula. All of us until now have considered Horton's evaporation equation as another in a long list of empirical equations available to estimate lake evaporation. Solomon Vimal and Vijay Singh have provided us with a forensic analysis of Horton's research, much of which is buried as unpublished material.*

**SV and VS:** We are privileged to have you as a reviewer, as your recent review paper on the subject of evaporation (McMahon et al, 2019) was a key reference for us. We thank the Handling Editor (Prof. Beven) for requesting your comments.

> *TM: The authors have discussed thoroughly each aspect of the vapour removal from a water surface  - diffusion, wind action  and convection – in relation to each component of Horton's lake evaporation formula. In doing so they have provided at least to this reviewer a unique explanation of the various evaporative processes that occur at or near the lake surface.*

*The paper addresses a key question in hydrology, and it is most appropriate that it be published in HESS. Not only is it novel, but it addresses an important hydrologic issue, the calculation of lake evaporation. The title of the paper reflects clearly the content and sufficient details are provided in the Abstract for a curious reader to be excited to read it.*

*Although long in length, the paper is clearly and concisely written.*

**SV and VS:** We are delighted to receive your positive feedback.

*TM: I do have a number of edits, mainly minor, which I list below. Because Horton worked in the US system of measurement, the discussion around numerical values is mainly in those units. I strongly recommend the authors include the metric equivalent values wherever possible especially with respect to key parameters and equations, for example, Equation 3b.*

**SV and VS:** Thank you for this comment. We will include metric equivalents in the Supplementary as part of a visual flowchart type schematic for practitioners, which also addresses a later comment of yours about outlining the steps to compute pan and lake evaporation in an easy to follow manner for practitioners. We will direct the reader to Supplementary in multiple places in the paper such that the reader will not miss it.

*TM: L72: "etc" is unhelpful. Please insert other contributions or delete.*

**SV and VS:** Thank you for pointing this out, we have deleted it.

*L84: It would be helpful for future researchers to include in the supplementary material not only the year and title of Horton's work but also where the material can be accessed.*

**SV and VS:** The paper links are not fully verified (some are broken), so it does not fit within our schedule to include the full reference with this paper, but we invite anyone interested to drop us an email for the current in-progress spreadsheet where citations and access details are available for many of Horton's papers.

Though we cannot at this time share an incomplete citation list, to aid the reader and to address the comment, we will include a section in Supplementary, titled, "**Tips to find Horton's papers and full bibliography:** 1) google this: "$title + $year + "Robert E. Horton" (side note: we found it easy to save the full citation using Zotero's plugin for browsers); 2) search in AGU's Virtual Hydrology bibliography list maintained here - https://connect.agu.org/hydrology/vhp-scope/roberthorton; 3) Check the online archive of Albion College (Horton's *Alma Mater*) ; 4) contact SV by email to check in his personal, unfinished, bibliography (access can be granted to an in-progress Google Sheet where notes on bibliography and the content and working website/download link are curated); 5) Go to National Archives in Maryland and dig into the 94 boxes (see list

of boxes in Beven, 2004a). One of these 5 approaches should help you access the full paper and citation.

*L102: I think the word "kettle" will be unfamiliar to many. May I suggest this be briefly explained or another term used.*

**SV and VS:** Thank you for pointing this out, we will include a brief explanation.

*L102: Please indicate the location of the Hemlock lake system.*

**SV and VS:** This is a good idea, we will do so.

*L113: Comment in parenthesis is incorrect. The citation to Horton was from Rohwer (1931) as noted in Table 1 of McMahon et al. (2016).*

**SV and VS:** Thank you for this comment. We cited Rohwer (1931), see in L59, but perhaps it is good to cite it again here.

*L305: The term evaporative capacity is used several times in Section 3.1 and in Section 3.3. I am confused by its use. (i) Is this a term used by Horton? If so, then that should be made clear in the presentation. (ii) While I appreciate it is defined clearly in Equation1(a), it is, in fact, the pan evaporation. Why introduce a new term? (iii) In L315, the term Evaporation capacity is used. Is there a subtle difference between "evaporation capacity" and "evaporative capacity"? Is one a function of Vw and the other a function of Va? (iv) The definition in L320 appears similar to potential evaporation.*

**SV and VS:** Thank you for noticing this. In L307, we noted the same point as (ii) in your comment, that, "Pan evaporation ($E_P$), which is the same as evaporative capacity from lake ($E_{Cw}$)". The reason behind using two terms to mean the same thing is that we wished to highlight that Horton used different terms to mean the same thing, so while reading his papers, the authors can bear this in mind to avoid any confusion. We will clarify this to avoid confusion in the three instances where we may expect readers to have the same confusion as you did (i.e. in L305, L315 and L320).

*L314, "... in Sec 3": But this line is in Section 3.1. It seems to be referring to itself.*

**SV and VS:** Thank you for noticing this. We will change it.

*L314, "We provide revised values in Sec. 3 (Table 3)": This paragraph refers only to constant C. There is only one value of C in Table 3.*

**SV and VS:** Thank you for noticing this. We will change it.

*L315: "w.r.t": Suggest this be spelt out, and elsewhere in the manuscript.*

**SV and VS:** Thank you for noticing this. We will change it.

> *L323, 324: To me, this sentence is particularly important and may not be appreciated by practitioners wishing to apply Horton's equation. To aid future applications, it would be very helpful if the authors were to add another section to the manuscript listing succinctly the steps in applying Horton's procedure to an evaporation pan and to small and large lakes.*

**SV and VS:** Thank you for this comment. We will include a simple visual schematic to show practitioners, and note this in the conclusion so interested readers/practitioners can have a quick guide rather than add more text. We think the right place for this might be the supplementary section as the text is already very long.

> *Ls432,433: Clumsy sentence, needs rephrasing.*

**SV and VS:** Thank you for noticing this. We will rephrase it as follows: "Similar important observations from experiments by various scientists (Dalton, Schübler, Soldner) have not been taken into consideration in modern mass-transfer formulations of evaporation.".

> *L438: Unclear what is meant by "... motivate the position ...".*

**SV and VS:** Thank you for this comment. To make it clear, we will reword this as follows: "the position of the term allows the equation to generalize for condensation".

> *Ls503,505: In Equation (4a), why introduce another variable $E_{cw}$ when it equals $E_p$, and thus $F = E_L/E_p$. By not introducing $E_{cw}$, the explanation would be less tortuous.*

**SV and VS:** Thank you for this comment. We believe this comment was also raised by the Editor. We pondered if we should simplify it, but we decided to retain it as is to have it be consistent with how Horton defined his variables. But to address your point, we can allude to this confusion directly in the paper in L503. 505, as well as the previous comment where you raised a similar point (L305, L315 and L320).

> *L514: "These relationships...". It's unclear which equations "These" refer to. Please clarify.*

**SV and VS:** Thank you for this comment. We clarified this in the text.

> *L593: Because Equation (8a) is the key equation in the paper, may I suggest the word 'lake' be inserted between "general" and "equation".*

**SV and VS:** Thank you for this comment.  Yes, we will do so.

*L594: Again, as Equation (8a) is the key equation, I recommend strongly that the suffixes be included. I had to go back through the text to ensure I understood which values of V and v were being referred to.*

**SV and VS:** Thank you for this comment. We will include the suffixes.

*L646: This sentence needs redrafting. What does "… various shapes…" mean?*

**SV and VS:** Thank you for this comment. We can remove "various shapes" with no loss of meaning we wanted to convey in the sentence. By "shapes" we implied that the equation has same set of variables (except Konstantinov) by just a different position of coefficients.

*L660, Tables 1 and 2: Although Horton's equation exhibits the smallest bias in all cases, nevertheless, the bias for say one day is ~+16%, which is large. Could the authors put this value in some context with the level of bias expected from procedures other than the empirical one discussed in the paper. I don't know how widely empirical procedures are currently used in practice compared with other non-empirical procedures.*

**SV and VS:** Thank you for this comment. We can contextualize the expected errors based on literature reference. Non-empirical ones, e.g. Penman-Monteith (combination equation), may potentially produce a smaller bias, but would still have room for improvement because it partly relies on the aerodynamic equation which is empirical, and here shown to be less accurate than Horton's equation.

*L683: Capitalize "h" in "How".*

**SV and VS:** Thank you for this comment. We will capitalize it.

*L693: "1,68,300" !!!*

**SV and VS:** Thank you for catching this. We will correct this number. It seems to an earlier estimation in an older draft which we had later changed.

*L699, Table3: (i) "H" should be "â" rather than "H" as the latter is used as Height in Equation 3. (ii) What is the time-step relevant to the H, K and C values. This comment applies to the other formulae, but it is less important to know that.*

**SV and VS:** Thank you for this comment. We will make this correction.

*L734: I'm unclear why P and Q are included in Figure 2. The paper is about Horton's contribution to E.*

*I suggest P and Q be deleted from the figure.*

**SV and VS:** Thank you for the comment. This figure serves a particular purpose - the difference in E between the various climate normal periods may be readily explained by the anomalies in P which this figure provides. It can also help view the variation of P, E and Q together as a closed water balance system at the scale of aggregation, which is interesting to see, even though it is what we expect to see.

**SV and VS:** Again, we thank you and the editor for the careful review and editorial work that improved the quality of this manuscript.

With best wishes,

Solomon Vimal and Vijay P. Singh

---

## Author Comment (AC2)

**Author Responses**

**Re-discovering Robert E. Horton's Lake Evaporation Formulae: New Directions for Evaporation Physics**

Solomon Vimal[1], Vijay P. Singh[2]

[1]Department of Geography, University of California, Los Angeles, CA, 90049, USA

[2]Department of Biological and Agricultural Engineering & Zachry Department of Civil and Environmental Engineering, Texas A&M University, College Station, Texas 77802-2117, USA

Correspondence to: Solomon Vimal (solomonvimal@ucla.edu)

Dear Ms. Jansen and Dr. Teuling,

Thank you for your valuable review comments. Prof. Singh and I reviewed your comments, and we provide our point-by-point response below.

For convenience of reading, we have indented your comments, reduced font size, italicized, and changed color to blue.

Thanks and best wishes,

Solomon Vimal and Vijay P. Singh

**Responses to Reviewer 2**

*The manuscript by Vimal and Singh brings back to our attention the century old Horton's lake evaporation formula. The authors give a thorough historical overview on how the formula was developed and how it relates to other evaporation methods of varying complexity. The authors show us that Horton's formula outperforms the other methods. I have appreciated reading the manuscript that has been written in a story-telling form including quotes of the original papers of Horton. This provides the reader a good overview and sense on how the authors have reconstructed how Horton's formula was developed and subsequently fell into oblivion. The authors managed to re-awaken the use of it by applying it on data from a subarctic Canadian catchment and found that the use of the variable vapor pressure deficit (VVPD) term introduced by Horton is of added value compared to the use of only VPD which is frequently used in other evaporation methods. In short, I have read the manuscript with great interest and I think it fits the special issue History of hydrology. My suggestion is to publish the manuscript with very minor revisions for which I provide feedback in my comments below.*

**SV and VS:** We are pleased to receive this comment, and we thank Ms. Jansen and Dr. Teuling for their prompt review.

*General comments*

*- Please, provide units when explaining the variables of equations for clarity. In some cases it is given (e.g. p.12 L.358), but in most not.*

**SV and VS:** Thank you for pointing this out, we provided units in the first instance when any new term appears, but not in all subsequent instances. However, for the sake of clarity, we can add units in each step in the supplementary section where we plan to add a one page summary of the steps to use Horton's equation for practitioners (for pan and lake applications), where we can include metric and US units for comprehensiveness and clarity. We decided to include this after a comment we received from Reviewer 1 (see our responses to Reviewer 1)

*- The order of the tables as they are mentioned in the text is the other way around of the appearance of the tables itself.*

**SV and VS:** We thank you for pointing this out, we have noted this, and will update the manuscript.

*Specific comments*

*- Is there a specific reason why the authors are using θ for temperature, instead of the commonly used T? To my knowledge θ is more commonly used to indicate potential temperature.*

**SV and VS:** We thank you for this question. We use $\theta$ because it is the notation that Horton used. Our goal in this paper was to encourage readers to revisit Horton's original work so we feel that adopting his notations would aid the reader who wishes to revisit Horton's paper(s). To disambiguate that $\theta$ is not potential temperature, we will explicitly say so in the first instance when it appears. Thanks for pointing this out!

*- p.11 L.315 and L.325; w.r.t. – don't write as abbreviation*

**SV and VS:** We thank you for noticing this.

*- p.14 L.422-425; in more recent past, there are many other studies that have found Dalton's method to work well. Especially in the oceanographic community it is widely used. The authors could refer to that as well for a bit of nuance.*

**SV and VS:** We thank you for bringing this to our attention. It might be a good idea to make this note as a passing remark. We can look for recent literature from the Oceanographic community, but if you happen to have any key reference or authors that

come to mind, especially any that demonstrate that Dalton's method works particularly better than other methods, we would appreciate it if you can send the reference to the corresponding author (solomon.vimal@gmial.com).

> *- P.16 L.490; the reference of Vimal and Mikuszeit, 2021, is not included in the reference list of the manuscript.*

**SV and VS:** We thank you for this comment, please see line 990.

> *- P.18 L.534; humidity and temperature gradients is probably referring to horizontal gradients.*

**SV and VS:** We thank you for this comment. Yes, and we have a few thoughts to share here (perhaps this can go unsaid in the manuscript): as we understand it, the humidity gradient can be both horizontal (due to dry wind moving moisture) and vertical (as vapor blanket thickness can vary substantially: the maximum approaches infinity for fully saturated air over a large lake) and temperature gradient referred here is horizontal (perhaps caused due to vertical mixing at various parts of the lake that have variable depths (bathymetry), shading effects from cloud, mountain, etc.).

> *- P.18 L.533/534; do the authors have a reference that underpins the statement that evaporation rates are constant over large lakes?*

**SV and VS:** Thank you for this question. This was not meant in any strict sense, and our thinking was as follows, and we hope this addresses your concerns (if any). We will clarify this thinking succinctly in the line you quoted: over a pan, the role of vapor blanket is important, but the contribution of variable evaporation in the fringes of the lakes is negligible (not to say vapor blanket height variation doesn't exist, but that variable evaporation rates can be ignored as the area, fringes of lakes, involved is small enough). If this sentence is read outside this paragraph, it has little meaning as over large lakes we can expect vertical mixing of lakes which changes the evaporation rates with or without vapor blankets, but in summers and in all cases of more than laminar (gentle, convective moisture transporting) surface wind (which is most of the time), and more generally, over lakes that are warmer than 4 degrees (if other factors that affect density and incoming radiation aren't significant), they would have more or less the same temperature because of how wind induced turbulent mixing happens rapidly on the surface and equalizes the temperature. But this equalization can be offset by vertical mixing in cold lakes.

> *- P.18 L.534-539; First, the authors state that horizontal variability of the thickness of the vapor blanket is negligible, while the next paragraph is*

*dedicated to the importance of horizontal variation and it is mentioned that this is the main theoretical breakthrough of Horton. Please, make this transition more clear or explain better.*

**SV and VS:** We thank you for this comment, we will introduce the following explanation to clarify this in the paper: "The reason for it being an important breakthrough is that it resolves and explains why pans and large lakes have different evaporation rates. It shows why in large lakes vapor blanket can be ignored, and why it would be a big mistake to ignore it from pans. This has large implications for the evaporation paradox."

*- P.20 L.583; typo: Vercauteeren --> Vercauteren*

**SV and VS:** Thank you for pointing this out!

*- P.20 L.584; do the authors mean 'were not explored' instead of 'were not unexplored'?*

**SV and VS:** Thank you for pointing this out! We will correct it.

**SV and VS:** Again, we thank both reviewers and the editor for the careful review and editorial work that improved the quality of this manuscript.

With best wishes,

Solomon Vimal and Vijay P. Singh

---

## Author Response (AR1)

**Responses to Editor's Comments**

**Re-discovering Robert E. Horton's Lake Evaporation Formulae: New Directions for Evaporation Physics**

Solomon Vimal[1], Vijay P. Singh[2]

[1]Department of Geography, University of California, Los Angeles, CA, 90049, USA
[2]Department of Biological and Agricultural Engineering & Zachry Department of Civil and Environmental Engineering, Texas A&M University, College Station, Texas 77802-2117, USA

*Correspondence to*: Solomon Vimal (solomonvimal@ucla.edu)

Dear Prof. Beven,

We thank you and the two reviewers for carefully reviewing our manuscript. A final list of changes we made that reflects our responses to the reviewer comments are provided below, together with a few additional minor changes.

Thanks and best wishes,

Solomon Vimal and Vijay P. Singh

**Point-by-point response to Reviewer's comments**

**Responses to Responses to Reviewer 1 (Dr. Thomas McMahon)**

> *TM: I am privileged to review this excellent article. The authors have provided an eclectic assessment of Robert E Horton's lake evaporation formula. All of us until now have considered Horton's evaporation equation as another in a long list of empirical equations available to estimate lake evaporation. Solomon Vimal and Vijay Singh have provided us with a forensic analysis of Horton's research, much of which is buried as unpublished material.*

**SV and VS:** We are privileged to have you as a reviewer as your review paper (McMahon et al, 2019) aided our work immensely. We thank the Handling Editor (Prof. Beven) for requesting your comments.

> *TM: The authors have discussed thoroughly each aspect of the vapour removal from a water surface - diffusion, wind action and convection – in relation to each component of Horton's lake evaporation formula. In doing so they have provided at least to this*

*reviewer a unique explanation of the various evaporative processes that occur at or near the lake surface.*

*The paper addresses a key question in hydrology, and it is most appropriate that it be published in HESS. Not only is it novel, but it addresses an important hydrologic issue, the calculation of lake evaporation. The title of the paper reflects clearly the content and sufficient details are provided in the Abstract for a curious reader to be excited to read it.*

*Although long in length, the paper is clearly and concisely written.*

**SV and VS:** We are delighted to receive your positive feedback.

*TM: I do have a number of edits, mainly minor, which I list below. Because Horton worked in the US system of measurement, the discussion around numerical values is mainly in those units. I strongly recommend the authors include the metric equivalent values wherever possible especially with respect to key parameters and equations, for example, Equation 3b.*

**SV and VS:** Thank you for this comment. We have included metric equivalents for equation 3c and equation 2d. The only other place where this may be needed is in Table 3, where the values of coefficients H, K and C are given, but the purpose of that table is not to demonstrate the formula, so we prefer to retain it as it is to avoid re-calibration of the 5 models. But we have included a simple guide (in the form of a table) in the appendix to help practitioners quickly apply Horton's formula in metric units.

*TM: L72: "etc" is unhelpful. Please insert other contributions or delete.*

**SV and VS:** Thank you for pointing this out, we edited it.

*L84: It would be helpful for future researchers to include in the supplementary material not only the year and title of Horton's work but also where the material can be accessed.*

**SV and VS:** We have included a section in the Supplementary to aid future researchers in their search for Horton's papers.

*L102: I think the word "kettle" will be unfamiliar to many. May I suggest this be briefly explained or another term used.*

**SV and VS:** Thank you for pointing this out, we have include a brief explanation, as follows: "kettle ponds (small ponds formed as a result of deglaciation)"

*L102: Please indicate the location of the Hemlock lake system.*

**SV and VS:** We have included this in L103: Rochester, New York.

> *L113: Comment in parenthesis is incorrect. The citation to Horton was from Rohwer (1931) as noted in Table 1 of McMahon et al. (2016).*

**SV and VS:** Thank you for this comment. We cited Rohwer (1931), see in L59, but it is good to cite it again here. We corrected it now.

> *L305: The term evaporative capacity is used several times in Section 3.1 and in Section 3.3. I am confused by its use. (i) Is this a term used by Horton? If so, then that should be made clear in the presentation. (ii) While I appreciate it is defined clearly in Equation1(a), it is, in fact, the pan evaporation. Why introduce a new term? (iii) In L315, the term Evaporation capacity is used. Is there a subtle difference between "evaporation capacity" and "evaporative capacity"? Is one a function of Vw and the other a function of Va? (iv) The definition in L320 appears similar to potential evaporation.*

> *L314, "... in Sec 3": But this line is in Section 3.1. It seems to be referring to itself.*

**SV and VS:** Thank you for noticing this. We have clarified this a bit to make it less ambiguous. We suppose that there is no subtle difference between "evaporative" and "evaporation" capacity. Our understanding is that Horton's evaporation papers were spaced apart by decades and he slightly varied his definitions in some cases, but we have tried to organize them into one place in a way that the ambiguous definitions he used are shown up front. We have given page level citation to the locations from where these definitions are taken in order to help the reader dig deeper if needed. Here are some changes we made:

The first definition, pan evaporation, according to Horton's definition, is the same as evaporation capacity from the lake, but pan evaporation is much more, i.e. it can be estimated with much more accuracy if taken together with the effects of vapor blanket. We have clarified this point by saying upfront that: *"Pan evaporation ($E_P$), used for first-order calculations, i.e. ignoring sub-pan variability of evaporation (see. Sec. 3.4), which is same as evaporative capacity referred to water surface temperature (notation used for this term elsewhere in Horton's work is $E_{Cw}$, e.g.* Horton, 1927, p. 160)."

We hope this is clearer than before.

> *L314, "We provide revised values in Sec. 3 (Table 3)": This paragraph refers only to constant C. There is only one value of C in Table 3.*

**SV and VS:** Thank you for noticing this. We changed it (values to value).

> *L315: "w.r.t": Suggest this be spelt out, and elsewhere in the manuscript.*

**SV and VS:** Thank you for noticing this. We have changed it.

> *L323, 324: To me, this sentence is particularly important and may not be appreciated by practitioners wishing to apply Horton's equation. To aid future applications, it would be very helpful if the authors were to add another section to the manuscript listing succinctly the steps in applying Horton's procedure to an evaporation pan and to small and large lakes.*

**SV and VS:** Thank you for this comment. We included a simple table in supplementary to show practitioners how to effectively use equations 1-6 for the three cases (lakes, small lakes, pans).

> *Ls432,433: Clumsy sentence, needs rephrasing.*

**SV and VS:** Thank you for noticing this. We rephrased this section. We hope it now reads better.

> *L438: Unclear what is meant by "… motivate the position …".*

**SV and VS:** Thank you for this comment. We reworded this to read better.

> *Ls503,505: In Equation (4a), why introduce another variable $E_{cw}$ when it equals $E_p$, and thus $F = E_L/E_p$. By not introducing $E_{cw}$, the explanation would be less tortuous.*

**SV and VS:** Thank you for this comment. We believe this comment was also raised by the Handling Editor. Though it does appear a bit tortuous. We added a point here to make it clear, "$E_p$ would be same as $E_{cw}$ if sub-pan variability is ignored and temperature of pan and lake are the same"

> *L514: "These relationships…". It's unclear which equations "These" refer to. Please clarify.*

**SV and VS:** Thank you for this comment. We clarified this by citing Horton (1927, p. 162)  when the area factor is introduced earlier in the section.

> *L593: Because Equation (8a) is the key equation in the paper, may I suggest the word 'lake' be inserted between "general" and "equation".*

**SV and VS:** Thank you for this comment.  Yes, we will do so. It reads "general lake or pan equation".

*L594: Again, as Equation (8a) is the key equation, I recommend strongly that the suffixes be included. I had to go back through the text to ensure I understood which values of V and v were being referred to.*

**SV and VS:** Thank you for this comment. We have included suffixes.

*L646: This sentence needs redrafting. What does "... various shapes..." mean?*

**SV and VS:** Thank you for this comment. We removed it to avoid confusion.

*L660, Tables 1 and 2: Although Horton's equation exhibits the smallest bias in all cases, nevertheless, the bias for say one day is ~+16%, which is large. Could the authors put this value in some context with the level of bias expected from procedures other than the empirical one discussed in the paper. I don't know how widely empirical procedures are currently used in practice compared with other non-empirical procedures.*

**SV and VS:** Thank you for this comment. We can contextualize the expected errors based on literature reference. Non-empirical ones, e.g. Penman-Monteith (combination equation), may potentially produce a smaller bias, but would still have room for improvement because it partly relies on the Aerodynamic equation which is empirical, and here shown to be less accurate than Horton's equation.

We added a few lines and a citation from literature: "This seems to also be true when considering other classes of models (radiation-based, temperature-based, combination) for open water evaporation, as evidenced by relative performances reported in Tan et al (2007). The only exception seems to be Artificial Neural Network (ANN) models which appear to have the potential to be marginally superior to Horton's, going by their relative performance, but they require sufficient site specific data and tuning."

*L683: Capitalize "h" in "How".*

**SV and VS:** Thank you for this comment. We will capitalize it.

*L693: "1,68,300" !!!*

**SV and VS:** Thank you for this comment. We will correct this number. It seems to be a typo from an earlier draft which was changed. Thank you for catching this.

*L699, Table3: (i) "H" should be "â□□" rather than "H" as the latter is used as Height in Equation 3. (ii) What is the time-step relevant to the H, K and C values. This comment applies to the other formulae, but it is less important to know that.*

**SV and VS:** Thank you for this comment. Please see line 699 (in the current version of the MS). The time-step relevant to H, K and C here is mm per 30 mins, as we used this time-step to compare all the parameters across all boot-strap samples at a common resolution (i.e. the resolution of the measured evaporation data).

> *L734: I'm unclear why P and Q are included in Figure 2. The paper is about Horton's contribution to E.*
>
> *I suggest P and Q be deleted from the figure.*

**SV and VS:** Thank you for the comment. We would like to retain this figure because it serves at least a couple of purposes: 1) it shows that the water balance at the scale considered here is largely closed, but not in all places; 2) it serves as a validation of E, i.e. the difference in E between the 4 time periods may be readily explained by the anomalies in P which this figure provides.

**End of review 1**
* * *
**Responses to Reviewer 2**

> ### *Review by Femke Jansen (supervised by Dr. Ryan Teuling)*
>
> *The manuscript by Vimal and Singh brings back to our attention the century old Horton's lake evaporation formula. The authors give a thorough historical overview on how the formula was developed and how it relates to other evaporation methods of varying complexity. The authors show us that Horton's formula outperforms the other methods. I have appreciated reading the manuscript that has been written in a story-telling form including quotes of the original papers of Horton. This provides the reader a good overview and sense on how the authors have reconstructed how Horton's formula was developed and subsequently fell into oblivion. The authors managed to re-awaken the use of it by applying it on data from a subarctic Canadian catchment and found that the use of the variable vapor pressure deficit (VVPD) term introduced by Horton is of added value compared to the use of only VPD which is frequently used in other evaporation methods. In short, I have read the manuscript with great interest and I think it fits the special issue History of hydrology. My suggestion is to publish the manuscript with very minor revisions for which I provide feedback in my comments below.*

**SV and VS:** We are pleased to receive this comment, and we thank Ms. Jansen and Dr. Teuling for their prompt review.

*- Please, provide units when explaining the variables of equations for clarity. In some cases it is given (e.g. p.12 L.358), but in most not.*

**SV and VS:** Thank you, we will do so. We added in L310. "He measured vapor pressure in inches of mercury and wind speed in miles per hour. Unless explicitly stated, for the purpose of illustration, these units will be used here. Metric equivalents are provided in the main text for equations where coefficients are introduced (also see Supplementary Section E)."

*- The order of the tables as they are mentioned in the text is the other way around of the appearance of the tables itself.*

**SV and VS:** Thank you for pointing this out. We updated this as, "Surprisingly, Horton's method outperforms other methods even when using estimated input vapor pressure (Table 2) even if the results of Horton's equation from Table 2 (estimated actual vapor pressure) are compared with the 5 methods from Table 1 (local measurements).". We hope this addresses your comment.

**Specific comments**

*- Is there a specific reason why the authors are using θ for temperature, instead of the commonly used T? To my knowledge θ is more commonly used to indicate potential temperature.*

**SV and VS:** Thank you for this question. We use $\theta$ because it was what Horton used. Our goal was to encourage readers to revisit Horton's original work, and we feel that adopting his notations would aid when one reads Horton's paper. To disambiguate that $\theta$ is not potential temperature, we will explicitly say so in the first instance when it appears. Thanks for pointing this out!

*- p.11 L.315 and L.325; w.r.t. – don't write as abbreviation*

**SV and VS:** We thank you for noticing this. This was already noted by the first reviewer and we have now fixed it.

*- p.14 L.422-425; in more recent past, there are many other studies that have found Dalton's method to work well. Especially in the oceanographic community it is widely used. The authors could refer to that as well for a bit of nuance.*

**SV and VS:** We thank you for bringing this to our attention. We did not find any suitable literature for Dalton's equation in the Oceanography literature. From our analysis it appears that Dalton's method is the least favorable.

*- P.16 L.490; the reference of Vimal and Mikuszeit, 2021, is not included in the reference list of the manuscript.*

**SV and VS:** We thank you for this comment, please see line 990 (of the preprint MS).

*- P.18 L.534; humidity and temperature gradients is probably referring to horizontal gradients.*

**SV and VS:** We thank you for this comment. As we understand it, the humidity gradient can be both horizontal (due to dry wind moving moisture) and vertical (as vapor blanket thickness can vary substantially, and the maximum approaches infinity for fully saturated air over a large lake) and temperature gradient referred here is horizontal (perhaps due to vertical mixing at various parts of the lake that have variable depths, shading effects from cloud, trees, mountain, etc.).

*- P.18 L.533/534; do the authors have a reference that underpins the statement that evaporation rates are constant over large lakes?*

**SV and VS:** Thank you for this question. This argument is made from a general understanding, and my thinking is as follows, and I hope this addresses your concerns: over a pan, the role of vapor blanket is important, but the contribution of vapor blanket over large lakes becomes negligible (not to say it doesn't exist, but that variable evaporation rates can be ignored as the area involved is small enough). If this sentence is read outside this paragraph, it has little meaning as over large lakes we can expect vertical mixing, but in summers and in all cases of more than laminar (gentle, convective moisture transporting) surface wind (which is most of the time), and more generally, over lakes that are warmer than 4 degrees (if other factors that affect density and incoming radiation aren't significant), they would have more or less the same temperature because of how wind induced turbulent mixing happens rapidly on the surface and equalizes the temperature. But this equalization can be offset by vertical mixing in cold lakes. After some thought, to avoid such a lengthy explanation, and to be more precise, we removed that line that evaporation is contact over large lakes.

*- P.18 L.534-539; First, the authors state that horizontal variability of the thickness of the vapor blanket is negligible, while the next paragraph is dedicated to the importance of horizontal variation and it is mentioned that this is the main theoretical breakthrough of Horton. Please, make this transition more clear or explain better.*

**SV and VS:** We thank you for this comment, we introduced an explanation to clarify this in the paper: "The reason for it being an important

breakthrough is that it resolves and explains why pans and large lakes have different evaporation rates. It shows why in large lakes vapor blanket can be ignored, and why it would be a big mistake to ignore it from pans. This has large implications for the evaporation paradox."

> *- P.20 L.583; typo: Vercauteeren --> Vercauteren*

**SV and VS:** Thank you for pointing this out!

> *- P.20 L.584; do the authors mean 'were not explored' instead of 'were not unexplored'?*

**SV and VS:** Thank you for pointing this out! We will correct it.

**End of review 2**

**List of further changes made to the revised manuscript**

1. Some 5 tips were added to Supplementary file to conduct an effective search to find Horton's papers and their full citations.

2. Section 3.2.2 "Adjustment of $\Psi$ for pan geometry" was re-arranged slightly to read better.

3. We made minor edits to Section 1.3

4. Added a missing reference: Millar, F. G. "Evaporation from Free Water Surfaces." Canada Dept. Transport, Div. Meteorol. Services, Canadian Meterol. Mem., Vol. 1. 2: 39–65, 1937.

5. Added a new reference: Tan, S. B. K., Shuy E. B., and Chua L. H. C.: Modelling Hourly and Daily Open-Water Evaporation Rates in Areas with an Equatorial Climate, Hydrological Processes 21, no. 4, 486–99. https://doi.org/10.1002/hyp.6251, 2007.

6. Removed an unreferenced citation: Walter, K. M., Zimov, S. A., Chanton, J. P., Verbyla, D. and Chapin, F. S.: Methane bubbling from Siberian thaw lakes as a positive feedback to climate warming, Nature, 443(7107), 71–75, doi:10.1038/nature05040, 2006.

7. Made a more conservative conclusion based on comments from Handling Editor.